# CINner: Modeling and simulation of chromosomal instability in cancer at single-cell resolution

**Khanh N. Dinh**[1,2*], **Ignacio Vázquez-García**[1,3,4,5*], **Andrew Chan**[6], **Rhea Malhotra**[3,7], **Adam Weiner**[3,8], **Andrew W. McPherson**[3], **Simon Tavaré**[1,2*]

**1** Irving Institute for Cancer Dynamics, Columbia University, New York, New York, United States of America, **2** Department of Statistics, Columbia University, New York, New York, United States of America, **3** Computational Oncology, Department of Epidemiology and Biostatistics, Memorial Sloan Kettering Cancer Center, New York, New York, United States of America, **4** Department of Pathology, Krantz Family Center for Cancer Research, Massachusetts General Hospital and Harvard Medical School, Boston, Massachusetts, United States of America, **5** Broad Institute of MIT and Harvard, Cambridge, Massachusetts, United States of America, **6** Case Western Reserve University, Cleveland, Ohio, United States of America, **7** Stanford University, Palo Alto, California, United States of America, **8** Tri-Institutional PhD Program in Computational Biology and Medicine, Weill Cornell Medicine, New York, New York, United States of America

* knd2127@columbia.edu (KND); ivazquez-garcia@mgh.harvard.edu (IV); st3193@columbia.edu (ST)

## Abstract

Cancer development is characterized by chromosomal instability, manifesting in frequent occurrences of different genomic alteration mechanisms ranging in extent and impact. Mathematical modeling can help evaluate the role of each mutational process during tumor progression, however existing frameworks can only capture certain aspects of chromosomal instability (CIN). We present CINner, a mathematical framework for modeling genomic diversity and selection during tumor evolution. The main advantage of CINner is its flexibility to incorporate many genomic events that directly impact cellular fitness, from driver gene mutations to copy number alterations (CNAs), including focal amplifications and deletions, missegregations and whole-genome duplication (WGD). We apply CINner to find chromosome-arm selection parameters that drive tumorigenesis in the absence of WGD in chromosomally stable cancer types from the Pan-Cancer Analysis of Whole Genomes (PCAWG, $n = 718$). We found that the selection parameters predict WGD prevalence among different chromosomally unstable tumors, hinting that the selective advantage of WGD cells hinges on their tolerance for aneuploidy and escape from nullisomy. Analysis of inference results using CINner across cancer types in The Cancer Genome Atlas ($n = 8207$) further reveals that the inferred selection parameters reflect the bias between tumor suppressor genes and oncogenes on specific genomic regions. Direct application of CINner to model the WGD proportion and fraction of genome altered (FGA) in PCAWG uncovers the increase in CNA probabilities associated with WGD in each cancer type. CINner can also be utilized to study chromosomally stable cancer types, by applying a selection model based on driver gene mutations and focal amplifications or deletions (chronic lymphocytic leukemia in PCAWG, $n = 95$). Finally, we used CINner to analyze the impact of CNA probabilities, chromosome selection parameters, tumor growth

**Data availability statement:** CINner is available as an R package at https://github.com/dinhngockhanh/CINner.

**Funding:** This work was funded in part by an Ovarian Cancer Research Alliance (OCRA) Ann Schreiber Mentored Investigator Award to IVG [650687]. AW is supported by NCI Ruth L. Kirschstein National Research Service Award for Predoctoral Fellows F31-CA271673. RM was supported by the Computational Biology Summer Program at Memorial Sloan Kettering Cancer Center, funded by an R25 grant from the National Institutes of Health [5R25CA233208-04]. The funders had no role in study design, data collection and analysis, decision to publish, or preparation of the manuscript.

**Competing interests:** The authors have declared that no competing interests exist.

dynamics and population size on cancer fitness and heterogeneity. We expect that CINner will provide a powerful modeling tool for the oncology community to quantify the impact of newly uncovered genomic alteration mechanisms on shaping tumor progression and adaptation.

## Author summary

Chromosomal instability (CIN) is a hallmark of cancer, characterized by the acquisition of structural and numerical chromosomal alterations in malignant cells. Toward understanding how CIN affects cancer evolution and cell fitness, it is necessary to integrate experimental and computational approaches that capture the temporal dynamics and consequences of CIN in tumor tissues. We present CINner, a framework for modeling CIN during cancer evolution. CINner is designed to output data that are compatible with both bulk and single-cell DNA sequencing methods, enabling the analysis of tumor heterogeneity and clonal evolution at different resolution levels. Application of CINner to bulk data reveals its ability to characterize specific cancer types with chromosome-arm selection parameters, which reflect the bias between tumor suppressor genes and oncogenes on those arms. CINner can also be used to model the increase in CIN associated with whole-genome duplication, a frequently observed and early event in many cancers. On the other hand, CINner can also study cancer types driven mainly by changes in specific genes. CINner is available as an R library, and we expect that it will provide a powerful modeling tool for the oncology community, toward quantifying the impact of genomic alterations on shaping tumor progression and adaptation.

## Introduction

Chromosomal instability (CIN) is a hallmark of cancer, characterized by the acquisition of structural and numerical chromosomal alterations in malignant cells. Key manifestations of chromosomal instability include chromosome missegregation, whole-genome doubling (WGD) and extrachromosomal DNA [1]. CIN generates genetic diversity and phenotypic variation among cancer cells, which can facilitate their adaptation to different environmental challenges, such as metastasis, drug resistance, and immune evasion [2]. On the other hand, CIN can also impair cell fitness by causing cellular stress, impaired DNA repair, and reduced proliferation. The role of CIN in cancer is therefore complex and context-dependent, and depends on the balance between its benefits and costs. To better understand how CIN affects cancer evolution and cell fitness, it is necessary to integrate experimental and computational approaches that can capture the temporal dynamics and consequences of CIN in tumor tissues.

We present CINner, a framework for modeling chromosomal instability during cancer evolution. CINner is designed to output data that are compatible with both bulk and single-cell DNA sequencing methods, enabling the analysis of tumor heterogeneity and clonal evolution at different levels of resolution. One of its advantages over existing algorithms is the ability to accommodate distinct copy number aberration (CNA) mechanisms that result from CIN and collectively transform a cell's karyotype and fitness. CINner uses a number of numerical techniques to enhance the speed and efficiency of the simulations. It can generate the clonal dynamics for cell populations of sizes comparable to real tumors, from which the phylogeny tree and cell-specific measurements can be simulated for a subsample of cells,

mimicking how DNA-sequencing data is produced. The framework allows for easy implementation of genomic events ranging in size from WGD to focal amplification/deletion and point mutations. The selection component of CINner is formulated as a function mapping a cell's karyotype and single nucleotide variants (SNVs) to its fitness. At one extreme, cell fitness can be defined solely upon the aneuploidy pattern, which is appropriate for studying certain solid tumors with prevalent widespread CNAs [3–5]. At the other extreme, CINner can model cancers that are mostly diploid and driven predominantly by recurrent point mutations and focal indices targeting specific driver genes [6–8]. As most cancers can be characterized as driven mainly by recurrent mutations or CNAs, or a mixture of both [9], CINner is uniquely positioned to uncover evolutionary patterns in many tumors. Finally, cancer cells in CINner evolve according to a stochastic branching process model, constrained by the carrying capacity of the environment. The tumor growth pattern even in the same cancer type can vary between exponential and logistic with a decades-long steady-state level, with implications for genetic composition, disease progression and clonal extent [10]. The carrying capacity model therefore provides the flexibility to examine the effects of the tumor dynamics on its heterogeneity and fitness.

## Results

### CINner models chromosomal instability during cancer evolution

In CINner, each cell is characterized by its copy number (CN) profile, or driver single nucleotide variant (SNV) profile, or both (Fig 1a). As genomic regions are amplified or deleted as copy number aberrations (CNAs) occur, the SNVs residing in those regions are correspondingly multiplied or lost. CINner models cancer evolution as a branching process [11]. Cell lifespan is exponentially distributed with an input turnover rate, similar to previous works [12,13]. At the end of its lifespan, the cell either divides or dies. This assumption is mathematically equivalent to other models such as [14], where cell division and death are simulated as two independent exponentially distributed processes [15]. The probability for a cell to divide depends on its fitness, determined by its CN and mutation profiles according to a selection model. The division probability is also calibrated so that the population size follows established dynamics. After a cell division, daughter cells either have the same profiles as the mother cell, or harbor CNA or driver SNVs events resulting in new profiles.

Previous mathematical models have mainly studied the evolution of SNVs during cancer development [16,17]. Some recent works have focused instead on analyzing the intra-tumor heterogeneity and convergence of CNAs [18,19]. CINner is distinct from most cancer evolution models in its ability to incorporate both SNVs and CNAs during cancer evolution and study how they impact the selection landscape simultaneously. SimClone [20] is another algorithm capable of generating synthetic tumor data with both genomic change classes. CINsim [21] is another method that allows modeling of CNAs in single cells and focuses on inferring rates of chromosome missegregation. However, unlike other methods, CINner can accommodate five distinct CNA mechanisms, each with distinct alteration patterns and varying impacts on cell fitness (Fig 1b). Whole-genome duplication (WGD) results in one daughter cell with double the genomic material of the mother cell. Whole-chromosome missegregation misplaces a chromosome strand among the two daughter cells. In contrast, only a strand arm is misplaced in chromosome-arm missegregation. Finally, focal amplification and deletion typically impact shorter subchromosomal region in a strand arm, and either increases the number of copies or deletes them in this region in one daughter cell.

Three selection models are included. The first model characterizes the selection of chromosome arms (Fig 1c), with the following assumptions:

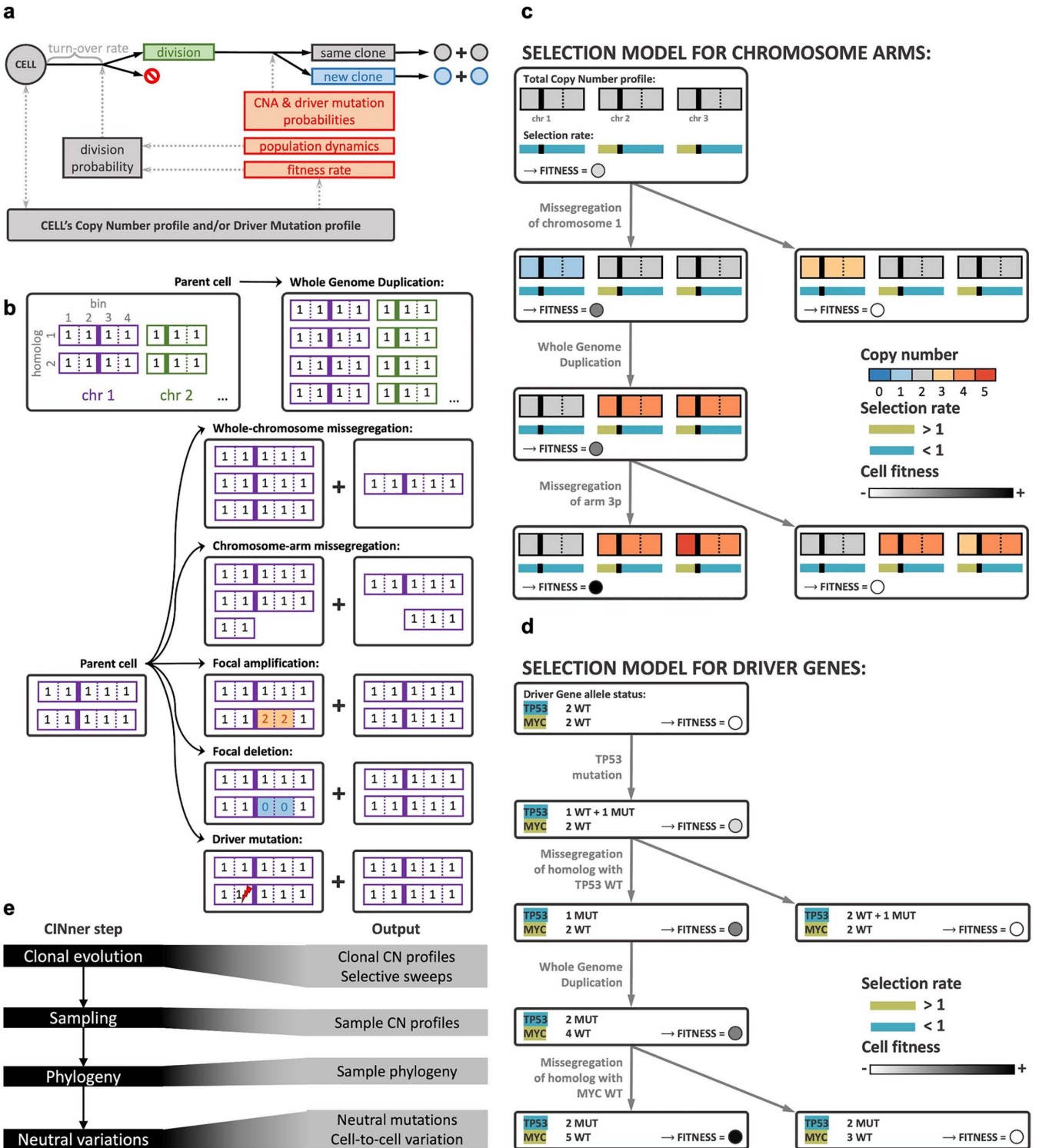

**Fig 1. Overview of CINner's mathematical model and simulation algorithm. (a)** Each cell is characterized by a copy number profile and/or driver mutation profile, which define its fitness rate. The lifespan for each cell is exponentially distributed with the same turnover rate. The probability of a cell to divide instead of dying takes into account its fitness rate and the population size, such that the population size follows known dynamics. If a cell divides, it can create new clones if a Copy Number Aberration (CNA) event occurs or a new driver mutation is acquired, otherwise the daughter cells belong to the same clone. **(b)** CNA and driver mutation events can occur during a cell division. Each chromosome homolog is represented as a vector, where each entry is the CN in a bin (vertical solid lines represent centrosomes, separating the two chromosome arms). Whole Genome Duplication results in one progeny with two copies of each homolog in the parent cell's genome. Other events are chromosome specific. During a whole-chromosome missegregation, one random

homolog is misplaced between the two progeny cells. During a chromosome-arm missegregation, a homolog is torn between the progeny, with one cell gaining a random arm and the other cell losing that arm. Focal amplification and deletion target a random region on a random chromosome arm, and either doubles the CN across all bins in that region (amplification, yellow bins) or resets the CN with 0 (deletion, blue bins). Driver mutation does not affect the CN profiles, but one allele of a randomly selected driver gene is changed from wild-type to mutant in a cell (lightning symbol). **(c and d)** Two selection models included in CINner. Squares represent cells, profiles of which change according to CNAs and driver mutations. Circles in each cell represent its fitness (darker is fitter). **(c)** Selection model for chromosome arms. A bin's total Copy Number is the sum of its CN across homologs. The selection rates, which are constant across cells, measure the total effect of genes on each arm (vertical solid lines represent centrosomes, separating arms). Arms dominated by Tumor Suppressor Genes (TSGs) have selection rates < 1 (blue), their losses increase the cell's fitness and gains decrease it (e.g., missegregation of chromosome 1 in this example). The opposite holds true for arms with selection rates > 1 (green), which house Oncogenes (OGs), gains of which increase the cell's fitness (e.g., missegregation of arm 3p) WGD does not change the arm balance and therefore the cell's fitness rate remains constant. **(d)** Selection model for driver genes. Each driver gene has a selection rate for its wild-type (WT) and mutant (MUT) alleles, which are constant across cells. The balance of all driver gene allele counts and their selection rates defines a cell's fitness rate. A cell has higher fitness rate if a TSG (blue) is either mutated or lost, or an OG (green) is either mutated or gained. Here TP53 represents a TSG and MYC represents an OG. A third hybrid selection model is a combination of **(c)** and **(d)**. All selection models are further subject to viability checkpoints. If a cell violates thresholds on nullisomy extent, maximum bin CN, driver counts, etc. then its fitness rate is zero and the cell eventually dies. **(e)** Left: schematics of the simulation algorithm, divided into four consecutive main steps. Right: data available to be computed in each step (see S1 Notes). CINner can complete prematurely if the later steps are not necessary, depending on the data requested by the user.

- For chromosome arms with selection parameter $s > 1$: gains increase the cell fitness and losses decrease fitness. This change increases with higher $s$.

- For arms with $s < 1$: losses increase fitness and gains decrease fitness. The impact increases with higher $1/s$.

The selection parameter serves as an indicator for the balance of tumor suppressor genes (TSGs) and oncogenes (OGs), as arms with high OG counts are commonly amplified and arms with many TSGs frequently get lost in cancer [22].

The model for selection of driver mutations (Fig 1d) seeks to portray the selection of individual TSGs and OGs directly. In this model, the selection parameters for the wild-type (WT) and mutant (MUT) alleles of a gene, are defined according to whether the gene functions as a TSG or an OG in that specific cancer type. We assume that a cell's fitness increases when

- A TSG is mutated or lost, or

- An OG is mutated or gained.

This model is based on the "one-hit" hypothesis [23], where each additional driver gene hit renders the cell more advantageous. The third model is a combination of these two models, describing cancer as driven both by small events targeting driver genes and large CNAs changing gene balance across the genome.

WGD is an early and ongoing event in many cancers [24,25], and is associated with an altered selection landscape [26] and chromosomal instability (CIN) [27,28]. In this manuscript, we focus on the increased CIN resulting from subsequent mitoses after a cell acquires WGD. Because all three models are defined upon the gene balance in a cell, a cell retains the same fitness immediately after WGD. However, we assume that the cell and its progeny have significantly higher CNA probabilities during division. Furthermore, each selection model is subject to viability checkpoints, which eliminate cells that exceed defined thresholds on driver mutation count, bin-level CN, average ploidy, or extent of nullisomy. S1 Notes describes the mathematical model in detail.

CINner is developed to efficiently simulate observed SNVs and CNAs in a tumor sample (Fig 1e). To optimize for computing memory and runtime, the genome is divided into bins of a fixed size, and the allele-specific bin-level copy number profile of each cell is tracked throughout tumor progression. Each new mutation is assigned a genomic location, and gets multiplied or deleted if the site is affected by later CNAs. Two observations are utilized to increase the efficiency of CINner. First, cells with the same phylogenetic origin share the

same CN and mutational profiles, therefore they evolve similarly throughout time. Second, the information relevant for downstream analysis is restricted to only the sampled cells. Therefore, it is not necessary to simulate single cells in the whole population individually, and instead we focus on clones, defined as groups of cells that have identical CN and mutational characteristics. The first step of CINner consists of simulating the evolution of clones in forward time. New clones are generated when CNAs or driver mutations occur, and the clone sizes change through time according to the branching process governing cell division and death. We use the tau-leaping algorithm [29] for efficiency, as the exact Gillespie algorithm [15] is time-consuming for cell populations of the typical size of tumors. In the second step, CINner samples cells from predefined time points. Next, it constructs the phylogeny for the sampled cells by using the "down-up-down" simulation technique [30]. In short, the sampled cell phylogeny is generated as a coalescent (cf. [31]), informed by the recorded clone-specific cell division counts throughout time from step 1. Finally, cell-to-cell variations due to neutral CNAs and passenger mutations are simulated on top of the phylogeny tree and trickle down to the sample observations. CINner is explained in more detail in S1 Notes.

The use of the "down-up-down" strategy and tau-leaping algorithm allows for significantly reduced runtime of CINner, compared to directly simulating the branching process for the whole population and then extracting the phylogeny only for the sampled cells. The runtime of simulating the clonal evolution (step 1, Fig 1e) scales with the number of clones and the number of time steps. The clone count increases with higher CNA and driver mutation probabilities. The number of time steps increases inversely with the step size selected for the tau-leaping algorithm. Meanwhile, the computational cost of simulating the phylogeny (step 3, Fig 1e) scales with the number of sampled cells, which is typically of magnitudes smaller than the population size.

We note that the data simulated in the forward and backward steps capture complementary views of cancer evolution. CINner's first step simulates the whole population throughout time. The population at each time point is characterized by distinct co-existing clones, their CN and driver mutation profiles, and their cell counts. This information contains all clones that arise during the whole process, including those that become extinct or are rare at the final time and hence unrepresented in the sample taken in step 2. Therefore, the output data allows for the examination of the expansion and/or extinction of any given clone in the simulation. On the other hand, the sample phylogeny from step 3 in CINner captures the history of a subsample of cells taken at the final time point. It therefore represents information that is observable from a hypothetical tumor biopsy. The phylogeny depicts recent subclonal evolution, but may lack (a) a full view of the heterogeneity in the whole cell population, and (b) information about early population genetic processes, e.g., before the sample MRCA. Depending on the applications, the users may utilize the data from either step for their analyses.

## Selection parameters calibrated for chromosome arms predict gene imbalance and prevalence of whole-genome duplication

We develop a parameter estimation program for the chromosome arm selection model (Fig 1c), which employs the Approximate Bayesian Computation random forest (ABC-rf) method [32] (Fig 2a and S1 Notes). We find that simultaneous parameter inference for both selection parameters and CNA probabilities in bulk DNA sequencing data results in nonidentifiability issues. Previous works have observed around $1-9 \times 10^{-3}$ missegregations per division in cancer cell lines [33–35]. However, in CINner this figure can be explained by either (i) high CNA probabilities coupled with selection parameters close to 1, or (ii) low CNA probabilities and selection parameters farther from 1. We will examine this in more detail in the next section.

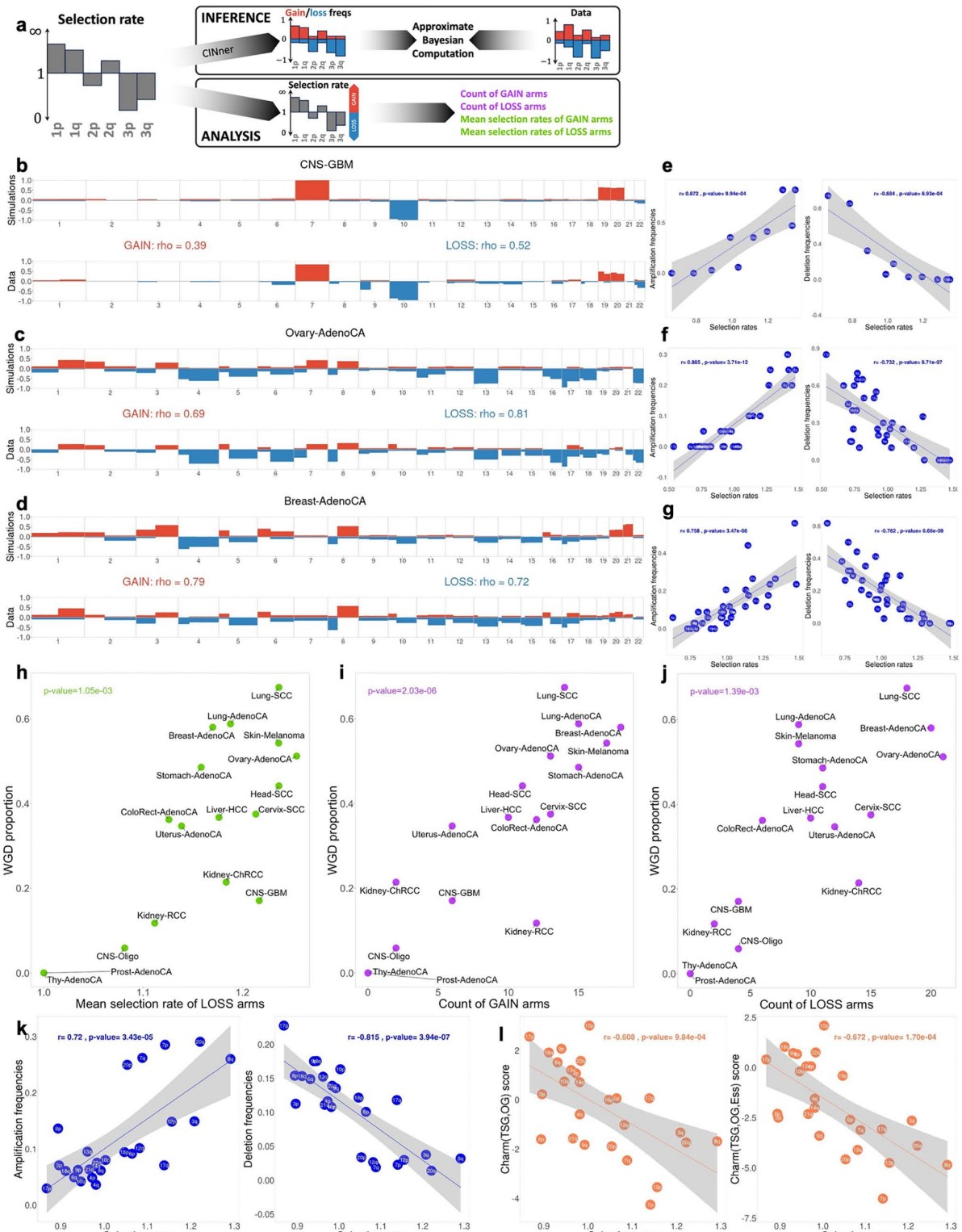

**Fig 2. Results from fitting the chromosome arm selection model to CN data from PCAWG and TCGA. (a)** Schematic for the inference and analysis of cancer type-specific chromosome-arm selection parameters. In the inference, selection rates are sampled from a prior distribution. CINner simulations are created for each parameter set, from which the gain/loss frequencies are computed across the genome. We then find the posterior distributions that match the frequencies observed in the data with Approximate Bayesian Computation. In CINner, gains of arms with selection rates > 1 are advantageous, similarly as losses of arms with selection rates < 1. Therefore, to analyze

the results, we classify each arm as a GAIN arm if mean posterior selection rate > 1, and as a LOSS arm if mean posterior selection rate < 1. The inference result for each data cohort can then be summarized with the count and mean selection rate of GAIN and LOSS arms. Read S1 Notes for more information. **(b–d)** Comparison between gain/loss frequencies from the fitted model (top) and non-WGD samples in PCAWG (bottom) for the cancer types diffuse glioma in central nervous system (**b**; CNS-GBM, $n = 34$), ovary adenocarcinoma (**c**; Ovary-AdenoCA, $n = 20$) and breast adenocarcinoma (**d**; Breast-AdenoCA, $n = 34$). Spearman's correlation coefficient rho between frequencies of gains (or losses) among all bins in PCAWG and simulations. **(e–g)** Correlation between inferred chromosome arm selection rates and amplification/deletion frequencies for individual chromosome arms in CNS-GBM (**e**), Ovary-AdenoCA (**f**) and Breast-AdenoCA (**g**). Linear regressions and p-values from Pearson correlation. **(h–j)** Correlation between WGD proportion and mean inverse selection rates of LOSS arms (**h**), and counts of GAIN (**i**) and LOSS arms (**j**) in each PCAWG cancer type. p-values from Spearman correlation between WGD proportion and each variable. **(k and l)** Fitted selection rates versus TCGA pan-cancer chromosome arm gain/loss frequencies (**k**) and gene balance scores (**l**) (Davoli et al. 2013, $n = 8207$) [22]. The score Charm(TSG-OG) considers the gene imbalance between TSGs and OGs, and Charm(TSG-OG-Ess) additionally examines Essential genes. Linear regressions and p-values from Pearson correlation. Sample sizes for each cancer type are listed in S1 Table.

In this section, we choose to study the selection parameters for each cancer type, and whether they indicate the tissue-specific selective pressure. Therefore, we fix the whole-chromosome missegregation probability at a comparatively low rate of $5 \times 10^{-5}$ for all cancer types, so it is easier to analyze the inferred selection parameters. Given a CN data cohort, we find $f_r^{[gain]}$ and $f_r^{[loss]}$, the frequencies of gain and loss for each chromosome arm. The ABC method then finds the posterior probability distributions for the arm selection parameters and the chromosome-arm missegregation probability that explain the observed gain and loss frequencies. We obtain a point estimate for each parameter by using its maximum a posteriori probability (MAP) estimate, which is the mode of the parameter's posterior distribution [36]. Finally, we generate CINner simulations with this estimated parameter set for direct comparison against the CNA data. The estimated chromosome-arm missegregations appear to be similar across different tumor types, possibly as a result of the nonidentifiability (S1–S17 Figs). Therefore, we focus the analysis on the inferred selection parameters.

We first employ the parameter fitting routine to study distinct cancer types with available data from Pan-Cancer Analysis of Whole Genomes (PCAWG) [37]. Samples with whole-genome duplication (WGD) have been shown to exhibit significantly different selection forces from non-WGD samples of the same cancer type, especially with respect to chromosome arm gains and losses [26]. Therefore, we limit the data to only non-WGD samples in each PCAWG data type for parameter calibration. We apply the parameter inference to 17 cancer types with $n \geq 10$ non-WGD samples. In the CINner framework, gains of a chromosome arm $r$ with selection parameter $s_r \gg 1$ are selective, therefore sequenced samples exhibit high $f_r^{[gain]}$. Conversely, chromosome arms with $s_r \ll 1$ exhibit high $f_r^{[loss]}$. On the other hand, arms with low $f_r^{[gain]}$ and $f_r^{[loss]}$ do not frequently get altered copy number, so we assume that these arms are neutral. Therefore, we limit the inference only to chromosome arms with $\left| f_r^{[gain]} - f_r^{[loss]} \right| \geq 0.1$, to mitigate the effects of neutral evolutionary noise.

Fig 2b presents the fitting results for glioblastoma samples (CNS-GBM). This cancer type has relatively few frequent CNAs, except for the combination of chromosome 7 gain and chromosome 10 loss [38], and gains of chromosomes 19 and 20 at lower frequencies [39]. Compared to CNS-GBM, ovarian adenocarcinoma (Ovary-AdenoCA) contains extensive CNAs shaped by multiple mutational processes [40], especially genomic loss-of-function events in *BRCA1* and *BRCA2* genes [41] (Fig 2c). Finally, breast adenocarcinoma (Breast-AdenoCA) is also associated with high aneuploidy [3] (Fig 2d). For each cancer type, the gain/loss frequencies produced from the simulator with fitted selection parameters closely resemble the genomic CNA landscape from PCAWG. Additionally, the selection parameters for individual chromosome arms correlate strongly with their amplification or deletion proportions (Fig 2e–g). Similarly, the inferred chromosome arm selection parameters for the other

14 cancer types lead to similar CNA landscapes to those from PCAWG (S1–S17 Figs). Overall, this demonstrates the model's ability to uncover specific selection forces characteristic of particular cancers, regardless of the extent of aneuploidy or bias toward genomic gains or losses.

We then examine whether the estimated parameters are indicative of cancer properties, specifically the selection for whole-genome duplication (WGD). It occurs in about 30% of tumors and is associated with a poor prognosis, suggesting that it plays a crucial role in cancer development [42]. WGD is also linked with extensive and profound changes in the selective landscape, including heightened chromosomal instability [43,44], increased preference for losses over gains [45], and changes in co-occurrence and mutual exclusivity in aneuploidy patterns [26]. Because of WGD's typical occurrence in initial stages of tumorigenesis and the many genomic changes it causes up to diagnosis [42], it is difficult to infer from DNA-sequencing data the causes for selection of WGD in early cancer development. We investigate whether the chromosome arm selection parameters inferred from CINner can predict tissue-specific WGD prevalence in PCAWG. We also explore which features correlate strongly with WGD proportion, which would imply contribution to increased fitness for WGD cells over the non-WGD population. For a given cancer type, we classify chromosome arms $r$ with $\left| f_r^{[gain]} - f_r^{[loss]} \right| \geq 0.1$ with inferred selection parameter $s_r > 1$ as GAIN arms, and those with $s_r < 1$ as LOSS arms. Each cancer type is then characterized by the counts of GAIN and LOSS arms, together with their respective mean selection parameters.

One hypothesis for the prevalence of WGD in cancer is that WGD provides redundant genes to buffer the deleterious effects of nullisomy [44]. The risk of nullisomy increases if the CNA probabilities are high or if there exist LOSS arms with selection parameters $s \ll 1$. Because our missegregation probabilities are similar across tissue types due to nonidentifiability, the hypothesis predicts that WGD is more frequently observed in cancers with highly selective LOSS arms. The correlation between WGD prevalence and mean selection parameters in LOSS arms inferred from CINner across cancer groups confirms this (Fig 2h). Our results therefore are in agreement with the assumption that WGD helps cancer cells mitigate the risk of nullisomy from repeated losses in specific genomic regions [45,46]. On the other hand, the counts of GAIN and LOSS arms indicate the proportion of the genome that is under selection for CNAs. The correlation between WGD proportion and the counts of either GAIN or LOSS arms (Fig 2i and 2j) is compatible with evidence that WGD is associated with chromosomal instability in cancer [44]. In conclusion, we have shown that the selective landscapes uncovered by CINner can predict tissue-specific WGD prevalence, indicating that the inferred selection parameters are biologically meaningful. Moreover, the cancer types with either (i) many GAIN and LOSS chromosome arms, or (ii) some LOSS arms with high selection parameters, are more likely to harbor WGD, indicating that selection for WGD in cancer development is driven by its role in helping tumors avoid nullisomy and tolerate aneuploidy.

Finally, we investigate if our classification of chromosome arms as GAIN or LOSS, and their selection parameters calibrated by the fitting routine, can reveal the genetic imbalances within the arms. We calibrate the model on frequencies of chromosome arm gains and losses from the pan-cancer data in The Cancer Genome Atlas (TCGA) [22] (S18 Fig). Similar to the fitting results for PCAWG cancer types, there is a strong correlation between estimated arm selection parameters and the frequencies of either amplification or deletion (Fig 2k). We then compare the fitted selection parameters to chromosome arm scores in [22]. For a given chromosome arm, the Charm (TSG, OG) score accounts for the count and potency of tumor suppressor genes (TSGs) and oncogenes (OGs). The score is higher for arms with higher count or increased potency of TSGs, and lower for arms more abundant with OGs. The second score Charm (TSG, OG, Ess) additionally considers essential genes (Ess), in the same manner as OGs. The selection parameters derived from our model correlate well with both scores (Fig

2l), and the negative correlation reflects the selection model (Fig 1c). Chromosome arms with selection parameters $s \gg 1$ are under intense selective pressure to get amplified, indicating that they harbor many important OGs, hence low Charm(TSG,OG) or Charm (TSG, OG, Ess), and the opposite holds for arms with $s \ll 1$. The correlation is stronger for Charm (TSG, OG, Ess) than Charm(TSG,OG), possibly signaling the relevance of essential genes in shaping the selective landscape during cancer evolution. Overall, we have shown that the chromosome arm selection parameters uncovered in our model are biologically significant, as they reflect the bias in distribution and potency toward either tumor suppressor genes or oncogenes and essential genes. Therefore, the model can play a role in estimating the driver gene landscape in specific cancer types. Driver genes are largely identified through their mutation frequencies, therefore the cohort size limits the sensitivity to which rare driver genes can be detected [47]. However, the selection parameters fitted in our model are an estimate of the combined effects of genes located on the same chromosome arm, including both commonly altered genes and those with minor contributions to tumor growth. Importantly, the selection parameter fitting routine only requires a small cohort of cancer samples (S1 Table).

## Impact of selection, copy number aberration mechanisms and growth dynamics in the chromosome arm selection model

CINner provides a framework to study different families of models and analyze the impact of model parameters on observable statistics of individual cancer samples. We have shown that specific cancer types exhibit a wide discrepancy in chromosome arm driver count, the potency of these arms, and the distribution of GAIN and LOSS arms among them, as has been previously reported extensively [9,20,22,26,37,48]. We now analyze how these statistics change with different selection parameters in CINner. In this parameter study, the selection parameters calibrated for the TCGA pan-cancer dataset (Figs 2k and l and S18) are denoted as scale $\times 1$. We then study the sample statistics when GAIN or LOSS chromosome arms increase their selection parameters (Fig 3a–c). In CINner's framework, amplifications of GAIN arms are more advantageous as the selection parameters of these arms increase. During simulated evolution, cancer cells with these amplifications acquire higher fitness rates and are more likely to expand. The increased preference for gains over losses hence leads to higher average ploidy in the sample (Fig 3b). Although the samples contain more clonal gains, the counts of subclonal gains and losses decrease, because of shorter elapsed time from Most Recent Common Ancestor (MRCA) to when the sample is taken (Fig 3c). Interestingly, the count of clonal losses also increases slightly, as deletions behave as hitchhikers to amplification drivers. Conversely, as LOSS chromosome arms are more selective, the clonal loss count increases significantly and the clonal gain count increases moderately, while the subclonal gain and loss counts decrease, resulting in lower average sample ploidy (Figs 3b and S19d). In both cases, the heightened competition means that subclones either expand quickly or become extinct, therefore the tumor sample exhibits fewer subclones, lower Shannon diversity index, and more recent Most Recent Common Ancestor (MRCA) (Figs 3a, 3c, and S19).

Another area of interest is the effects associated with variable CNA probabilities on sample statistics. In particular, we study how different probabilities of missegregation impact cell fitness and tumor clonality (Figs 3d–f and S20). Although increasing selection parameters lead to heightened competition and therefore fewer subclones, a higher missegregation probability increases subclonal diversity and results in a larger Shannon diversity index (Figs 3d and S20a). Because of the enhanced diversity, subclones share more clonal gains and losses, and they also harbor more subclonal missegregations (Fig 3f). Because there is no change in the selection landscape, the MRCA age does not change appreciably. As a consequence, the ratio

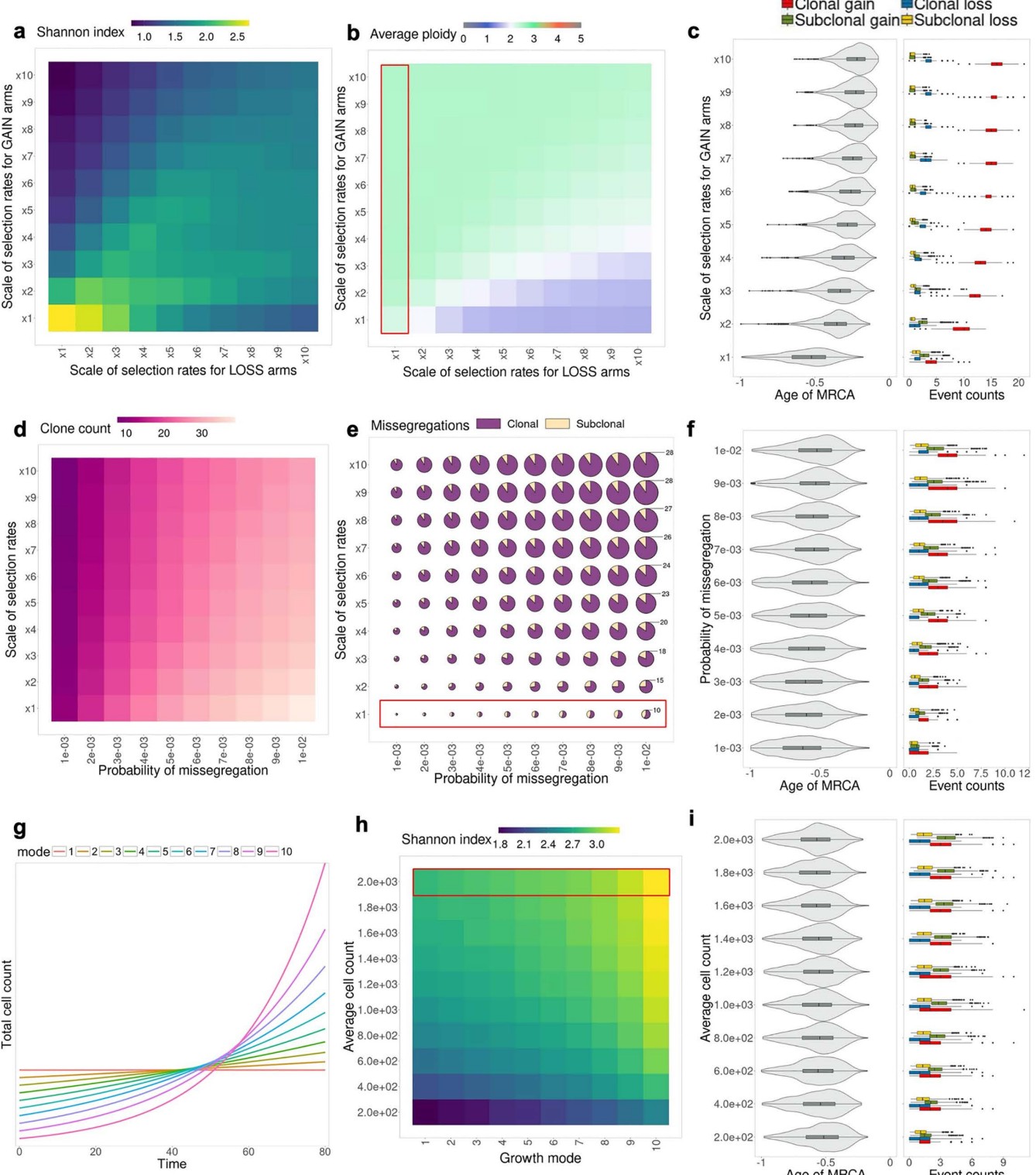

**Fig 3. Analysis of parameters in the chromosome arm selection model. (a and b)** Effects of selection rates for GAIN and LOSS chromosome arms on average Shannon diversity index (**a**) and average ploidy in each sample (**b**). (**c**) MRCA age and average missegregation counts, grouped based on clonal-ity (clonal/subclonal) and type (gain/loss), as selection rates for GAIN arms increase (variables correspond to highlighted segment in (**b**)). MRCA age is computed as fraction over time back to when simulation starts. MRCA age $= -1$ if the sampled cell phylogeny tree starts branching at the beginning of

the simulation. MRCA age $\rightarrow 0$ as the MRCA is closer to sampling time. (**d and e**) Effects of probability of missegregation and chromosome-arm selection rates on clone count (**d**) and average count of clonal and subclonal missegregations (**e**) (size of circles indicates the total missegregation counts). (**f**) MRCA age and missegregation counts as probability of missegregation increases (variables correspond to highlighted segment in (**e**)). (**g**) Different patterns of growth mode, ranging from constant (mode 1) to exponential with high growth rate (mode 10). (**h**) Effect of growth mode and average cell count on average Shannon diversity index. (**i**) MRCA age and missegregation counts as average cell count increases (variables correspond to highlighted segment in (**h**)). 1,000 simulations are created for every parameter combination.

of clonal to subclonal gain and loss counts remains constant as probability of missegregation varies (Fig 3e). This contrasts with increasing selection parameters, which likewise increases total missegregation count in the sample but with a higher bias toward clonal events. An important aspect to consider is that the total CNA count is the primary measure of CIN that can be obtained from bulk DNA-sequencing samples. However, as demonstrated in this study for missegregations, these counts can be explained by a spectrum of parameters in CINner, ranging from high CNA probabilities with low selective pressure (bottom right corner in Fig 3e) to high selection parameters coupled with low CNA occurrences (top left corner). As discussed in the previous section, it is therefore challenging to estimate both CNA probabilities and selection parameters with bulk DNA data.

Finally, we investigate how different growth patterns impact the cancer sample statistics, taking advantage of the model's ability to incorporate the dynamics of expected population size as input (Fig 3g–i). The cell turnover rate is unchanged under different tumor dynamics, hence the distribution of cell lifespan is constant. The sample size is also fixed at 1,000 cells for each parameter set. Despite this, as the population increases in size, the sampled cells both share more clonal missegregations and accrue more subclonal events (Fig 3i), leading to higher Shannon diversity index (Fig 3h). We also study ten different tumor growth models, ranging from constant to exponential with increasing growth rates (Fig 3g). In tumors growing at a low rate, the competition for carrying capacity is more intense. In contrast, higher exponential growth rate represents faster growing tumors, in which cells do not have to compete as much for space. As a result, even subclones with low fitness can expand, leading to higher clone count and Shannon diversity index (Figs 3h and S21).

In conclusion, the different components of the model, ranging from CNA mechanisms to tumor dynamics to selection model, have distinct effects on common cancer sample statistics. These signals are important to analyze, as they directly affect the process of model calibration. When using CINner to estimate parameters for specific datasets, it is crucial to find values for model constants that conform to the corresponding tissue type and tumor growth.

## The role of whole-genome duplication in promoting chromosomal instability

In previous sections, we estimated cancer type-specific selection parameters and missegregation probabilities in cell populations without whole-genome duplication (WGD) (Fig 2). WGD is a common and early event in many cancers [24], and is associated with an altered selection landscape [26] and heightened chromosomal instability (CIN) [27,28]. As has been observed in our parameter study, increasing CNA probabilities leads to higher heterogeneity, and the increased clonal competition results in greater tumor fitness (Fig 3d–f). In this section, we utilize CINner to measure the CIN level associated with WGD in a cancer-specific context.

The WGD proportion varies substantially among different cancers (Fig 2h–j). Moreover, the fraction of genome altered (FGA) among WGD and non-WGD samples in PCAWG further varies between cancer types (Fig 4a). For instance, although WGD samples of squamous

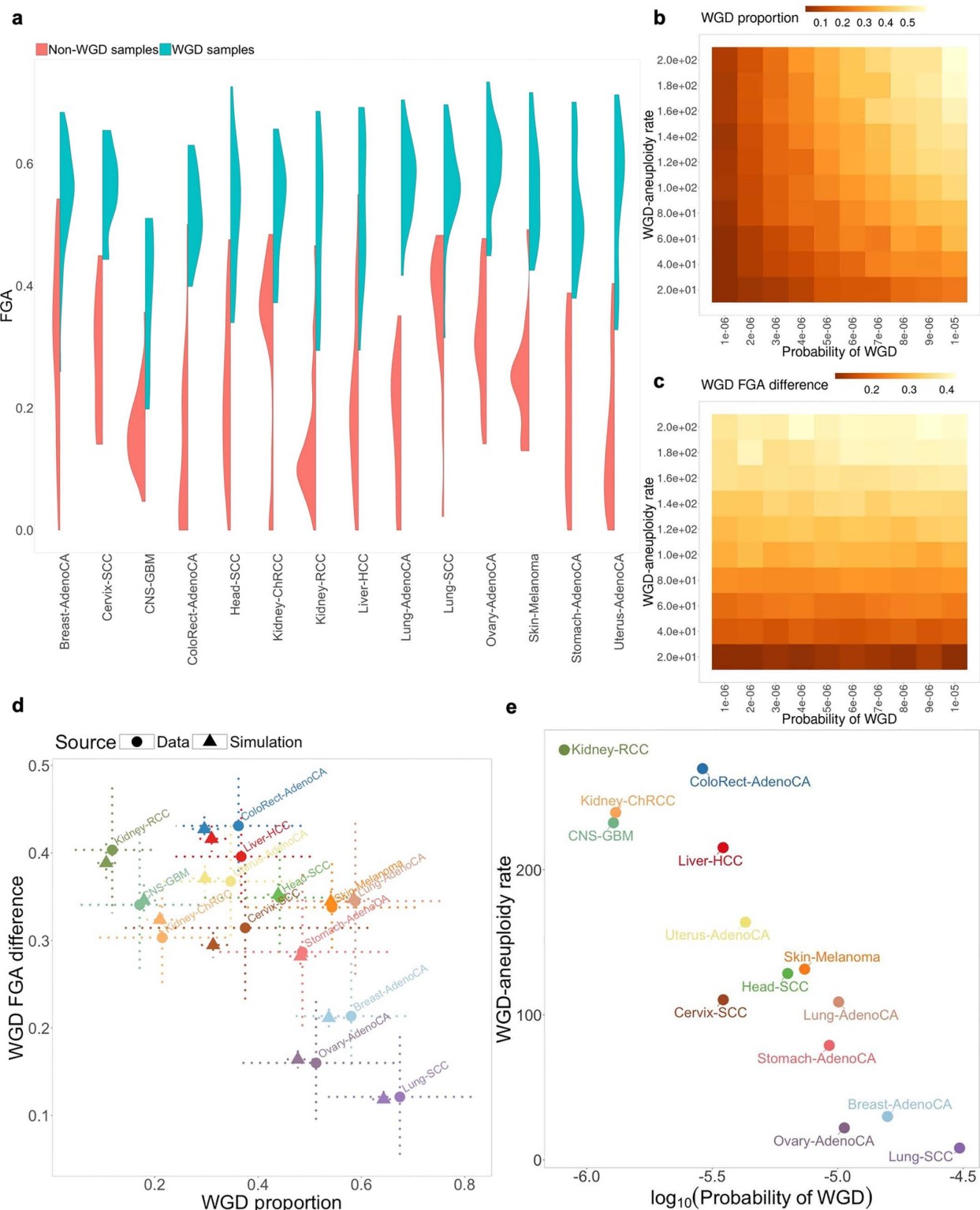

**Fig 4. Results from fitting WGD parameters to CN data from PCAWG. (a)** Distribution of Fraction of Genome Altered (FGA) in PCAWG by cancer type and WGD status. **(b and c)** Impact of varying WGD probability and WGD-aneuploidy rate on WGD proportion (**b**) and WGD FGA difference (**c**) in simulated samples. WGD FGA difference is defined as mean (FGA|WGD) - mean (FGA|non-WGD). 1,000 simulations are created for every parameter combination. **(d)** Comparison between WGD proportion and WGD FGA difference from the fitted model (triangle) and

PCAWG (circle). Dotted bars represent ranges of each statistic from 10,000 bootstrap samples. **(e)** Comparison between inferred WGD probability (in logscale) and WGD-aneuploidy rate for each cancer type. Sample sizes for each cancer type are listed in S1 Table.

cell lung carcinoma (Lung-SCC) exhibit highly altered genomes, the FGA of these samples is not much higher compared to non-WGD samples. In contrast, kidney renal cell carcinoma (Kidney-RCC) has few genomic alterations and therefore low FGA in non-WGD samples, however the WGD tumors exhibit high FGA, indicating significantly increased CIN level associated with WGD. We already captured the genome alteration landscape in non-WGD cancers (Fig 2). Therefore, in order to study the WGD-associated CIN, we characterize each PCAWG cancer type by two statistics: WGD proportion, and WGD FGA difference (defined as the mean difference in FGA between WGD and non-WGD tumors).

We define two parameters in CINner: the probability of WGD in each cell division, and WGD-aneuploidy rate, defined as the ratio of missegregation probabilities between WGD cells and non-WGD cells. We then study the changes in WGD statistics as these parameters vary, using the selection parameters fitted for the pan-cancer TCGA model (Fig 2k, 2l, and S18). As can be expected, increasing WGD probability leads to higher WGD proportion within the simulated samples (Fig 4b), however, the FGA difference between WGD and non-WGD samples is unchanged (Fig 4c). On the other hand, increasing WGD-aneuploidy rate raises the FGA difference, as WGD-cells accumulate missegregations at a higher rate. The higher heterogeneity within WGD cells also leads to emergence of karyotypes with higher fitness compared to non-WGD cells, ultimately resulting in higher WGD proportion (Fig 4b), similar to our observations from varying missegregation probabilities (Fig 3d–f).

We now infer the WGD probability and WGD-aneuploidy rate for distinct cancer types from the WGD proportion and FGA difference in each PCAWG cohort. For each cancer type, we assume the chromosome-arm selection parameters and missegregation probabilities inferred previously (Fig 2), then infer the WGD probability and WGD-aneuploidy rate with ABC-rf. To avoid overfitting, we limit the inference to 14 cancer types with at least 10 non-WGD samples and WGD proportion $>10\%$ in PCAWG. The posterior probabilities are largely unimodal (S22 and S23 Figs), indicating low uncertainty in the ABC inference. We simulated the WGD proportion and FGA difference using the modes for each inferred parameter, and the statistics are consistent with each PCAWG cohort (Fig 4d).

The comparison of inferred parameters between different cancer types reveals that the WGD probability per cell division and WGD-aneuploidy rate are negatively correlated (Fig 4e). One possible explanation is that there is a limit to the level of aneuploidy that can be tolerated in cancer cells, even in the presence of WGD. In cancer types with high WGD probability, there is a larger time span from WGD to diagnosis, as the event would frequently occur early in tumorigenesis. This results in increased aneuploidy, but also a large number of extreme karyotypes that are unviable. Therefore, the observed WGD samples exhibit much lower FGA as compared to expectations from CINner. Indeed, the FGA in WGD samples are more uniform across cancer types compared to the non-WGD samples (Fig 4a). Another explanation is that the increased FGA in WGD samples results from increased likelihood of multipolar divisions [49]. The resulting progeny exhibit highly aneuploid genomes, and most are nonviable due to nullisomy. However, it is possible that rare surviving cells are more selectively advantageous than diploid cells, and expand across the tumor. Under this assumption, the WGD-aneuploidy rate would be lower, as the WGD cells would already have markedly higher FGA after multipolar divisions.

We also compare the inferred probability of WGD against the average fraction of monosomy in diploid samples from PCAWG. Under the hypothesis that WGD helps cancer escape

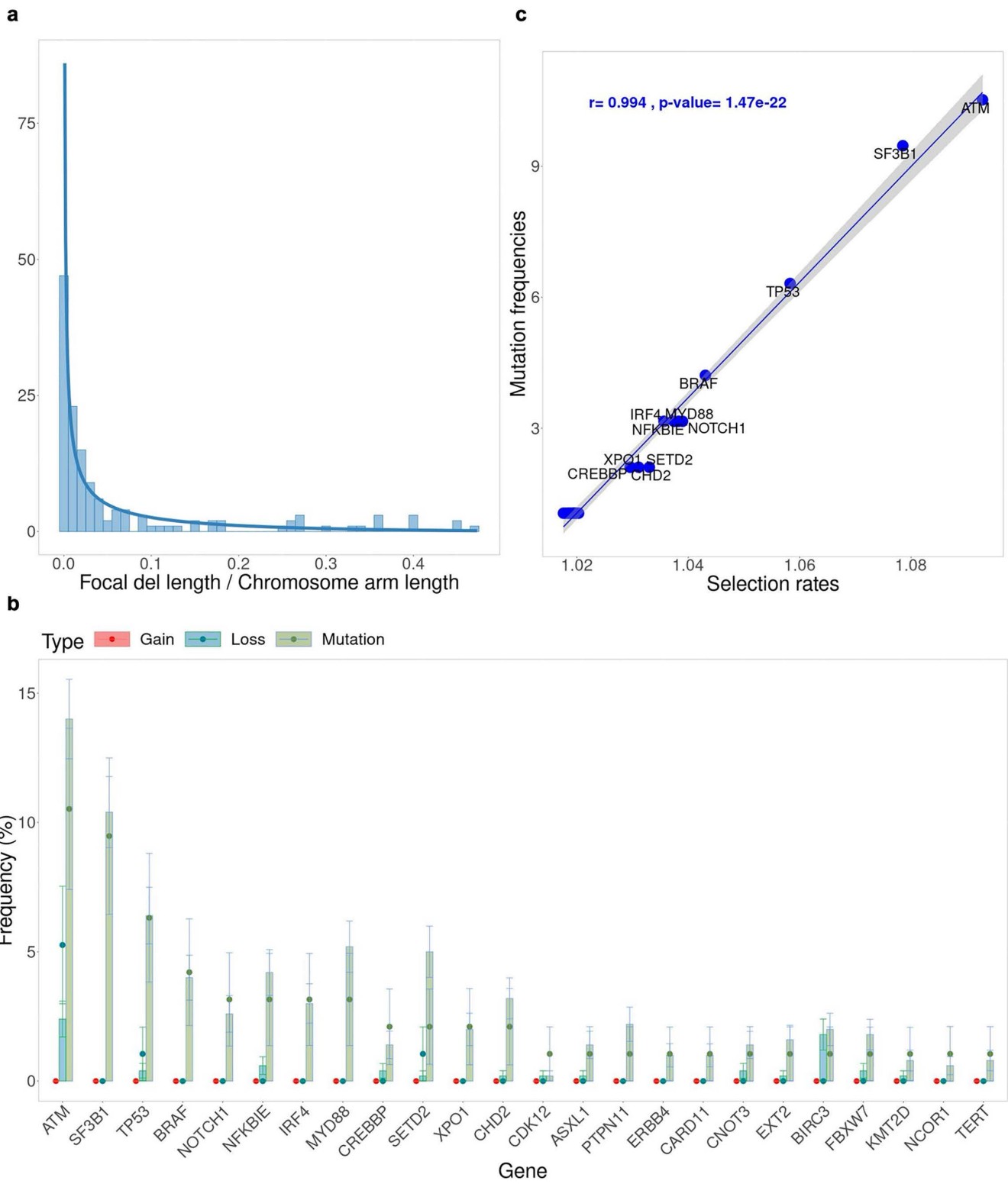

**Fig 5. Results from fitting the driver gene selection model to the CLLE-ES cohort in PCAWG ($n = 95$).** (a) Ratios of focal deletion lengths over corresponding chromosome arm lengths are fitted with a Beta distribution. (b) Comparison between mutation/gain/loss frequencies among driver genes from the fitted model (bars) and PCAWG (circles). Error bars represent the standard deviations from 10,000 bootstrap samples. (c) Correlation between inferred selection rates and mutation frequencies for individual driver genes. Linear regressions and p-value from Pearson correlation.

nullisomy, we would expect a higher WGD probability in cancer types that have higher monosomy fraction in diploid samples. The comparison is unclear (S23g and h Fig). One potential reason is that there are few samples for certain cancer types in PCAWG. The approach that we employ necessitates further subdividing the samples based on WGD status, which could render the statistics too prone to noise. However, the three cancer types with highest monosomy fraction, namely cervical squamous cell carcinoma (Cervix-SCC), breast and lung adenocarcinomas (Breast-AdenoCA, Lung-AdenoCA), have medium to high inferred WGD probability, compared to other cancer types. This might indicate that WGD indeed provides cancer a means to escape from nullisomy.

## Inferring selection parameters for driver genes in chromosomally stable tumors

Thus far we have examined the implementation of CINner in elucidating the selective roles of different CNA mechanisms, such as whole-chromosome and chromosome-arm missegregations and whole-genome duplication. Although these CNAs are frequently observed in cancer, many blood cancers and certain solid tumors are not associated with large-scale aneuploidies [9]. Here, we explore potential applications for CINner in analyzing such cancer types.

As a case study, we investigate the CLLE-ES cohort in PCAWG, consisting of chronic lymphocytic leukemia samples. The samples are largely diploid in general, therefore previously excluded in our chromosomal CNA study (Fig 2). We hypothesize that the cancer type is driven mostly by mutations and focal gains and losses, suggesting that our selection model for driver genes is appropriate (Fig 1d).

The list of driver genes is derived from tabulating all genes that are either mutated or impacted by CNAs in at least one CLLE-ES sample. Our selection model requires that each driver gene is assigned as either TSG or oncogene, under the assumption that losses of TSGs and gains of oncogenes are associated with increased fitness during tumorigenesis. Therefore, the driver genes are labeled as TSG or oncogene depending on whether the loss or gain frequency in the CLLE-ES cohort is higher, respectively. If the frequencies are equal, the driver gene role is taken from Cancer Gene Census [50]. We restrict the driver gene list to those that are listed in Cancer Gene Census, located on autosomes, and can be assigned a selective role.

We then model the lengths of focal amplification and deletion events separately, under the assumption that the ratio of a focal event length over the chromosome-arm length follows the Beta distribution (Fig 5a). Because none of the driver genes in our list is affected by focal amplifications (Fig 5b), we limit the parameter inference to driver mutation rate, focal deletion probability, and driver gene selection parameters. Similar to the CNA probability inference problem where the confounding effect of missegregation probabilities and chromosome-arm selection parameters results in nonidentifiability, here we fix the driver mutation rate and infer the other parameters relative to this value.

All posterior distributions of the driver gene selection parameters are more concentrated than their prior distributions and are unimodal (S24 Fig), confirming our ability to estimate the selection parameters effectively. Choosing the mode from each parameter's posterior distribution, we compare the driver gene event frequencies from CINner against the PCAWG data (Fig 5b). The mutation frequencies from CINner recover the signals from data. Moreover, the gene loss frequencies are largely in agreement with PCAWG observations. The inferred selection parameters exhibit a linear relationship with the mutation frequencies (Fig 5c), similar to the correlation between CINner-inferred chromosome-arm selection parameters and frequencies of amplifications and deletions for cancer types driven by CNAs (Fig 2e–g).

We note that currently the selection model is based on several simplifying assumptions. First, we assume haploinsufficiency for TSGs, i.e., cancer fitness increases with each additional allele inactivated via mutation or CN loss. For some driver genes, inactivation has been observed to require functional loss of both alleles (two-hit paradigm), and for yet other driver genes, the loss of one allele induces cancer but complete functional loss of the gene is toxic [23]. Second, our fitness formula is multiplicative, wherein each additional functional loss of a TSG or functional gain of an oncogene increases cell fitness by the same factor, dictated by the gene's selection parameter. Third, in some cancer types, the fitness associated with a de novo gene alteration depends on the genomic landscape, indicating the importance of the order in which driver gene mutations and CNAs occur. This phenomenon is not included in our selection model. Although these assumptions greatly reduce the complexity of the parameter inference while still largely capturing the gene alteration patterns in CLLE-ES, applications in other data cohorts and cancer types might necessitate a more involved selection model.

## Discussion

Cancer is characterized by a multistep process resulting in the reprogramming of key cellular components [51]. The genomic alterations that drive tumor heterogeneity and evolution range in extent and potency, from point mutations and small indels [52], to copy number aberrations and structural variants affecting one or multiple chromosomes simultaneously [43,51–53]. The occurrence rates of these distinct mutational processes and how they impact the selection landscape are highly context-dependent [43]. Loss or mutation of TP53 and BRCA1/2 leads to progressive increases in CNA rates and therefore tumor heterogeneity [53]. WGD likewise propagates chromosomal instability (CIN) [54], yet tolerance of WGD itself often also requires functional loss of TP53 [46,55]. Different alleles of a TSG can be deactivated via mutation, deletion [56], or copy-neutral loss of heterozygosity [57]. However, the same mechanisms can constrain each other, as shown for example by *MDM4* amplifications in the 1q arm, which are known to inhibit p53 signaling and accelerate tumor progression [58].

Because the effects of CNAs on cancer cell fitness depend heavily on context, it is challenging to fully understand the functional role of chromosomal instability from genomic measurements alone. Mathematical modeling can help distinguish between the effects of selection and neutral drift in CIN evolution, reveal how preference for specific karyotypes shapes cancer evolution, and forecast clonal dynamics [16]. Moreover, if both the occurrence rate and selection parameter of a CNA can be ascertained, we can infer the allele age and reconstruct the sample coalescence [59], which can offer valuable information in diagnosis and treatment selection.

To provide a comprehensive picture of how CIN impacts tumor progression, a model must possess two important features. First, it should account for the opposing forces of diversity and selection. Second, it should allow for the coexistence and interaction of different mutation and CNA mechanisms. Most models so far have only addressed some of these aspects. Early works studied chromosome copy number changes without selection, only assuming that nullisomic karyotypes were nonviable [60,61]. A recent model defined different phases of CIN in tumor growth, using breakpoint tally to measure subclone fitness [62]. Since copy number is not explicitly defined, this approach seems unsuitable for analyzing selection for optimal karyotypes. Another CIN study [63] employs CINSignatureGenomeSimulation [64], an algorithm to simulate the effects of different copy number signatures on the cancer genome, without direct modeling of selection. Other methods first generate a cell genealogy, then simulate CNAs along the branches [65,66]. However, the CNAs in this approach do not affect the phylogeny tree, therefore karyotype selection is not explicitly depicted. Some recent studies model the effects of selection and missegregation on subclonal copy numbers [18,67],

also incorporating point mutations [13] or WGD [28,68]. Nevertheless, many of these models focus only on the average ploidy and do not consider chromosome-specific CN [13,28,67]. In contrast, two studies directly integrate chromosomal selection parameters and study the resulting CN trajectories [18,68]. They employ existing chromosomal selection parameters defined from counts and potencies of OGs and TSGs from pan-cancer studies [22], to uncover the prevalent karyotypic trajectories during tumorigenesis. This seems to be the most promising approach to study the role of heterogeneity and selection in cancer on the copy number level. However, a potential drawback is that identification of cancer driver genes is nontrivial and its sensitivity depends on the sample size as well as the gene's mutation frequency, among other factors [69]. Therefore, although defining selection parameters based on known OGs and TSGs can give accurate results in a pan-cancer context, the approach seems to have limited applicability for studies of specific cancer types and datasets.

We present CINner, a model for simulating CNA mechanisms and selection in tumor evolution. It can accommodate various genomic events ranging in extent and impact, from point mutations to WGD. CINner uses several numerical techniques to reduce the memory and computing requirements of simulating whole genomes in large cancer populations. We use CINner to find chromosome-arm selection parameters from diploid PCAWG samples. The CN profiles simulated with the inferred parameters match the observed cancer-specific aneuploidy patterns. The estimated selection parameters predict WGD prevalence and correlate with driver gene count and potency in pan-cancer TCGA data. These signals indicate that the selection parameters inferred from CINner reflect the oncogenic effects of genes on specific genomic regions. Therefore, CINner can play an important role in modeling rare cancers, where driver gene identification is limited due to small sample size and low gene alteration frequencies. For these cancer types, we can instead infer the chromosome arm selection parameters with significantly fewer samples (Figs 2b–d and S1–S17). Each arm's selection parameter can then serve as an indicator for the combined impact of driver genes located on it. We also perform parameter analysis studies to quantify the effects of CNA probabilities, selection parameters, tissue cell count and tumor growth dynamics on CINner statistics. Finally, we apply CINner to cancers driven mainly by driver gene changes, such as CLLE-ES in PCAWG. In short, CINner is capable of modeling both small alterations impacting important genes and large-scale CNAs during tumor development.

An interesting finding from our parameter inference is that the WGD prevalence of a cancer type is connected to its chromosomal selection parameters in the diploid setting. This provides some insights into why WGD is a common early event in tumors, despite strong negative selection in normal tissues. In particular, WGD proportion correlates with higher selection parameters of TSG-acting chromosome arms (Fig 2h). As deletions of these arms are strongly selective, the cancer cells gradually lose copies due to missegregations, thereby risking the toxic effect of nullisomy. WGD could help alleviate this risk by raising those chromosome copy numbers above 1. Another explanation is that, because of the increased ploidy level, tetraploid cells can have repeated losses of a chromosome, resulting in higher impact on the gene balance. The WGD proportion also correlates strongly with the count of chromosome arms acting as either TSGs or oncogenes (Fig 2i and j). This could be because different cancer and tissue types have different levels of tolerance for aneuploidy and WGD. Alternatively, WGD has been shown to promote chromosomal instability [70]. The tetraploid cells, therefore, can explore the aneuploidy landscape and increase their fitness at a faster rate than diploid cells.

Although CINner has the power to study clonal dynamics at the single-cell level, the parameters were estimated by comparing pseudo-bulk simulations to bulk DNA sequencing data. This is to take advantage of the large sample sizes available with PCAWG and other cancer studies, to reduce the risk of overfitting. However, as shown in our parameter studies (Fig 3e), the chromosome gain and loss frequencies in bulk samples are similarly impacted

by CNA probabilities and selection parameters. This makes it challenging to infer both CNA probabilities and selection parameters simultaneously. Therefore, in this work, we fix the missegregation probabilities and focus on finding cancer-specific selection parameters.

Recently, technological advances in single-cell DNA sequencing have led to better resolution in capturing genomic data, and have demonstrated that tumors exhibit different levels of heterogeneity and chromosomal instability [53,62,71,72]. Our parameter studies show that single-cell statistics have different trends under variable CNA probabilities and selection parameters in CINner (Fig 3d–f). While higher missegregation probabilities result in increasing aneuploidy and sample diversity, higher selection parameters increase aneuploidy but decrease clone count, as a result of heightened subclonal competition. Recently, we studied the application of CINner to single-cell data in a synthetic setting [73]. We considered a wide range of statistics based on observed CN profiles and inferred phylogeny trees from single-cell data. We found that the ABC framework can recover accurate posterior distributions not only for chromosome-specific selection parameters but also missegregation probabilities. We plan to combine this inference framework with CINner simulations to analyze currently available single-cell data [25,53,72]. As single-cell data increases in sample size and cell count, CINner can be implemented to estimate both the occurrence rate and fitness impact of different CNA mechanisms, circumventing the nonidentifiability issues in bulk data.

CINner's capabilities are compared against previous algorithms in S2 Table. Although CINner provides the framework to model, simulate and infer parameters from cancer type-specific data for a wide range of CNA mechanisms, there are more CIN processes that will benefit from future works. Chromosomal rearrangements, whether in specific loci or in chromoplexy and chromothripsis events, have been frequently observed across different cancers [37,74]. However, there is a lack of mathematical models for the evolution of cells resulting from these events during tumorigenesis. This is likely because defining the fitness rates for these cells is particularly challenging, given the large number of possible genomic configurations and the lack of understanding how the relocated genes impact cell selection. Another potential area for modeling is how the amplifications and deletions of specific alleles might impact the selection landscape differently.

In conclusion, we have shown that CINner offers a comprehensive framework to analyze the interplay between selection and distinct genomic alteration mechanisms. CINner can simulate individual cells and clones in a sample, making it adaptable for DNA data ranging from single-cell to bulk level. Its flexibility can accommodate data from different DNA-seq technologies, including targeted sequencing [75,76], and enable incorporation of new CNA mechanisms or point mutations [77]. We envision that with the advent of large genomic studies using both bulk and single-cell approaches, CINner will enable accurate parameterization of cancer evolution.

## Methods

### Characterization of cells and clones in CINner

In CINner, the genome is divided into bins of equal length and we assume each bin in each cell has one allele configuration. Let $N$ be the chromosome count, $L$ be the length of each bin and $M$ be the number of bins spanning the whole genome. Let $\mathcal{M}_i$ be the bin count in chromosome $i$, whose centromere is in bin $\mathcal{C}_i$. We then have $\mathcal{M} = \sum \mathcal{M}_i$ and $\mathcal{L} \cdot \sum \mathcal{M}_i$ is the total length of the genome. For typical single-cell data, $\mathcal{N} = 24$, $\mathcal{L} = 500,000$ bp and $\mathcal{M} = 6206$. The copy number (CN) information of a cell is then characterized by:

- $\{J_i : i = 1,\dots,\mathcal{N}\}$ is the global CN count vector, where $J_i$ is the number of strands of chromosome $i$. $J_i$ changes if the cell gains or loses chromosome strands, e.g., via Whole Genome Duplication or chromosome missegregations.

- $\left\{K_{i,j}:i=1,\ldots,\mathcal{N};j=1,\ldots,J_i\right\}$ stores the local CN counts, where $K_{i,j}\in\mathbb{N}^{\mathcal{M}_i}$ is a vector of CN in each bin for strand $j$ of chromosome $i$. The entry $K_{i,j}(l)$ can increase or decrease if the cell undergoes amplifications or deletions affecting the bin $l$.

  For a diploid cell, $J_i=2$ and $K_{i,j}=(1,\ldots,1)$ in autosomes.
  Given the CN set up, every mutation in the cell is assigned an address $(a_1,a_2,a_3,a_4)$, where

- $a_1\in\{1,\ldots,\mathcal{N}\}$ denotes the chromosome

- $a_2\in\{1,\ldots,J_{a_1}\}$ denotes the homolog

- $a_3\in\{1,\ldots,\mathcal{M}_{a_1}\}$ denotes the bin along the homolog

- $a_4\in\{1,\ldots,K_{a_1,a_2}(a_3)\}$ denotes the CN unit in the bin

  A cell $k$ is characterized by its CN profile $\left(\left\{J_i^{[k]}\right\},\left\{K_{i,j}^{[k]}\right\}\right)$ and/or driver mutations $\left\{\left(a_1^z,a_2^z,a_3^z,a_4^z\right)\right\}_{z=1,\ldots,n^{[k]}}$, where $n^{[k]}$ is the total number of driver mutations that it has accumulated.

## Selection models

Given the profile of a cell $k$, we now define its fitness $s_k$. We explore three different models, of selection for chromosome arms, or driver mutations, or both. Each selection model is further subject to viability checkpoints.

**Selection model for chromosome arms.** In this model, we assume a library of chromosome arms $\{1p,1q,2p,2q,\ldots\}$, where each chromosome arm $r$ has selection rate $\lambda_r\in(0,\infty)$ and spans from bin $l_a^{[r]}$ to $l_b^{[r]}$. Given a cell $k$ with CN profile $\left(\left\{J_i^{[k]}\right\},\left\{K_{i,j}^{[k]}\right\}\right)$, we first find its ploidy:

$c^{[k]}=\dfrac{\sum_{i,j}K_{i,j}^{[k]}}{\mathcal{M}}$, rounded to nearest integer and the CN of each arm $r$ on chromosome $i_r$:

$c_r^{[k]}=\dfrac{\sum_{j;l=l_a^{[r]},\ldots,l_b^{[r]}}K_{i_r,j}^{[k]}(l)}{l_b^{[r]}-l_a^{[r]}+1}$, rounded to nearest integer.

The cell's fitness rate is then

$$s_k=\prod\lambda_r^{c_r^{[k]}/c^{[k]}}$$

Cells with gains of arms $r$ where $\lambda_r>1$ or losses of arms $r$ where $\lambda_r<1$ have increased fitness rates and therefore will expand. Conversely, losses of arms $r$ with $\lambda_r>1$ or gains of arms $r$ with $\lambda_r<1$ decrease the cells' fitness.

**Selection model for driver mutations.** This selection model assumes a library of driver genes for a specific cancer type. Each driver gene $d$ is located at bin $a_3^{[d]}$ on chromosome $a_1^{[d]}$ and has selection rate $\lambda_d\in(1,\infty)$, which varies for different genes. Based on the gene's specific role in the cancer type, we define its wild-type (WT) allele's selection rate $\lambda_{d-WT}$ and mutant allele's selection rate $\lambda_{d-MUT}$:

- If driver gene $d$ behaves as an oncogene (OG), then

$$\lambda_{d-WT}=\lambda_d$$

$$\lambda_{d-MUT}=\lambda_d^2$$

- If driver gene $d$ behaves as a Tumor Suppressor Gene (TSG), then

$$\lambda_{d-WT} = 1/\lambda_d$$

$$\lambda_{d-MUT} = 1$$

Given a cell $k$, from its CN and driver mutation profiles, CINner computes the CN for each driver gene. Specifically, let $n_{d-WT}^{[k]}$ and $n_{d-MUT}^{[k]}$ be the numbers of WT and mutant alleles of driver gene $d$, respectively. The cell's fitness rate is then

$$s_k = \prod \lambda_{d-WT}^{2 \cdot n_{d-WT}^{[k]}/c^{[k]}} \cdot \lambda_{d-MUT}^{2 \cdot n_{d-MUT}^{[k]}/c^{[k]}}$$

The formulae for driver gene-specific selection rates are based on some general assumptions about how cell fitness is affected by CNA and mutation events targeting individual driver genes:

i) WT alleles of oncogenes are less selective than mutant alleles which confer gains of function.

ii) Furthermore, amplifications of both mutant and WT oncogene alleles are selective.

iii) Mutations causing loss of function in TSGs are more selective than WT alleles.

iv) Deletions of TSGs are also selective.

v) However, once TSG alleles lose their function, they become neutral and their amplifications or deletions do not impact cell fitness.

**Hybrid selection model for chromosome arms and driver mutations.** The cell's fitness rate in this model is the product of fitness rates defined in the previous two sections. Therefore, cell's fitness depends both on CNA events affecting whole chromosomes, chromosome arms or genomes, and point events targeting driver genes.

**Viability checkpoints.** We include conditions for cell viabilities. If a cell $k$ violates any of these conditions, then its selection rate $s_k$ becomes 0 and the cell will die:

- Average ploidy: $c^{[k]} \leq \text{ploidy}_{max}$

- Highest bin CN: $\max_l K_{i,j}^{[k]}(l) \leq \text{CN}_{max}$

- Highest normalized bin CN: $\max_l \dfrac{K_{i,j}^{[k]}(l)}{c^{[k]}} \leq \text{CN}_{max}^{nor}$

- Nullisomy bin count: $\sum 1_{l:\sum K_{i,j}^{[k]}(l)=0} \leq \text{nullisomy}_{max}$

- Count of distinct mutated drivers: $\sum 1_{d:n_{d-MUT}>0} \leq \text{driver}_{max}$

- Count of WGD $\leq \text{WGD}_{max}$

## Cell evolution as a birth-death process

Cells in the population follow a birth-death process, with two properties defining a given cell $k$:

- The lifespan of the cell is exponentially distributed with rate $\lambda_k$, after which it either divides or dies. We assume $\lambda_k = \lambda$ is constant and equal to the turn-over rate of the cells, estimated from the literature.

- The cell divides with probability $p_k^{div}$ and dies with probability $1 - p_k^{div}$. If the cell divides, CNA and/or driver mutation events occur at predefined probabilities, in which case the profiles of the progeny cells are updated accordingly (see Section 1.3–1.4 in S1 Notes for details).

The division probability for cell $k$ with fitness rate $s_k$ at time $t$ is:

$$p_k^{div}(t) = g(t) \cdot f(s_k)$$

where $g(t)$ is a negative feedback loop ensuring that the current total cell count $P(t)$ follows a predefined population dynamics $\bar{P}(t)$ observed from either the data or the literature:

$$g(t) = \frac{\bar{P}(t)}{\bar{P}(t) + P(t)}$$

and $f(s_k)$ models the selection for the fittest cells:

$$f(s_k) = \frac{s_k \cdot P(t)}{\sum_{k'=1,\ldots,P(t)} s_{k'}}$$

## CINner's simulation algorithm

At initial time $t_0$, a CINner simulation starts with $\mathcal{N}_0$ clones. Each clone $n_0$ consists of $N^{[n_0]}$ cells with CN profile $\left( \left\{ J_i^{[n_0]} \right\}, \left\{ K_{i,j}^{[n_0]} \right\} \right)$ and/or driver mutations $\left\{ \left( a_1^z, a_2^z, a_3^z, a_4^z \right) \right\}_{z=1,\ldots,n^{[n_0]}}$. The simulation then follows four main steps (Fig 1e):

- Step 1 (clonal evolution): At each time point, CINner tracks the number of clones in the population, each clone's genotype, and the number of cells in each clone. The system is updated every time a new clone arises due to CNA or driver mutation events. The available data from this step consists of the clonal information, including the CN profile of each clone, its parent clone, the time it was born, and its cell count at each subsequent time point. The clones with increasing cell counts through time represent selective sweeps.

- Step 2 (sampling): A subsample of cells present at the final time is chosen. From this step, the CN profile of each sampled cell is available to the user.

- Step 3 (sampled cell phylogeny): The phylogeny for sampled cells in Step 2 are simulated in backward time, using information of clonal divisions in Step 1. This phylogeny tree is then available as output.

- Step 4 (neutral variations): neutral CNAs and passenger mutations are simulated and imposed on the sampled cell phylogeny from Step 3, without affecting clonal fitness rates. The profile of each cell, containing additional neutral CNAs and mutations, is then available for output.

  See Section 2 in S1 Notes for details.

## Inference for chromosome arm selection parameters

We retrieve pan-cancer data in TCGA from [22], and data for individual cancer types in PCAWG from https://dcc.icgc.org/releases/PCAWG/. To infer chromosome arm selection parameters, we extract samples without WGD in each cohort. Datasets with less than 10 non-WGD samples are not analyzed.

For each cohort, we find the gain/loss frequencies for individual arms. We then infer selection parameters using the Approximate Bayesian Computation (ABC) framework. From prior distributions

- Missegregation probability $p_{misseg} = 5 \cdot 10^{-5}$

- Arm-missegregation probability $p_{arm-misseg} \sim \text{Uniform}\left(10^{-5}, 10^{-4}\right)$
- Selection rates $\lambda_r \sim \text{Uniform}\left(0.5, 1.5\right)$

We create 10,000 CINner simulations. We then use abcrf [32], an ABC method in R that employs the random forest methodology, to infer the posterior distributions for $p_{arm-misseg}$ and $\lambda_r$ 's from the cohort's gain/loss frequencies. See Section 3 in S1 Notes for details.

### Inference for whole-genome duplication

In this section, we seek to estimate $p_{WGD}$, the probability that WGD occurs in each cell division, and $\alpha$, the WGD-associated CIN rate defined as

$$\alpha = \frac{p_{misseg}^{homolog;WGD}}{p_{misseg}^{homolog;non-WGD}}$$

Where $p_{misseg}^{homolog;non-WGD}$ and $p_{misseg}^{homolog;WGD}$ are the probability that a chromosome homolog is missegregation during a division of a non-WGD or WGD cell, respectively.

For each PCAWG type, we extract samples with WGD. Cancer types with WGD proportion $\leq 10\%$ are not analyzed. The posterior distributions for $p_{WGD}$ and $\alpha$ are inferred using abcrf, from prior distributions

$$\alpha \sim \text{Uniform}\left(0, 300\right)$$

$$\log_{10}\left(p_{WGD}\right) \sim \text{Uniform}\left(-6.5, -3.5\right)$$

See Section 5 in S1 Notes for details.

### Inference for driver gene parameters

We focus on CLLE-ES, which was excluded from our arm selection parameter and WGD inference due to largely diploid karyotypes. Distributions for the lengths of focal amplification and deletion events were fitted with Beta distributions to data. For each driver gene, we then compute the frequencies of gains, losses and mutations across all samples. The selection parameters for each driver gene are then inferred using abcrf, similar to previous sections. See Section 6 in S1 Notes for details.

### Supporting information

**S1 Notes. Details about CINner's mathematical model and simulation algorithm, inference method for missegregation and chromosome-arm selection parameters, parameter studies of the chromosome arm selection model.** Inference method for WGD parameters and driver gene parameters.
(PDF)

**S1 Table. Number of samples per PCAWG cancer type in our inference.** Estimation of chromosome-arm selection parameters (Figs 2 and S1–S17) was performed with non-WGD samples. Estimation of WGD probability and WGD-associated CIN (Figs 4, S22 and S23) employed WGD samples. *: Cancer types with WGD proportion ≤0.1 were excluded from the WGD inference.
(DOCX)

**S2 Table. Comparison of CINner's capabilities against currently available algorithms.**
Green indicates properties of cell-specific fitness, CNA mechanisms or aspects of the mutational process that an algorithm incorporates. Red indicates the algorithm does not include such properties. Yellow indicates properties that are included, with important caveats. †: CN breakpoint (loci where CN changes between genomic regions) can be directly computed from simulated CN profiles. *: cell fitness is modeled based on arm-specific selection coefficients computed from pan-cancer data [22]. ‡: CN profiles are input from observed data and not simulated. A: inference for mouse T-cell lymphoma and human colon cancer organoid single-cell data. B: inference for breast cancer. C: inference for yeast.
(DOCX)

**S1 Fig. Inference of chromosome-arm selection rates in Breast-AdenoCA (PCAWG, $n = 34$ ).** (**a**) Prior distribution (light blue) and posterior distribution (dark blue) from inference with ABC random forest. Broken line represents the mode in the posterior distribution for each parameter. (**b**) Comparison between simulations with fitted parameter (top) and gain/loss frequencies at arm level from TCGA (bottom). The simulations are computed with the posterior modes from (**a**). Spearman's correlation coefficient rho between frequencies of gains (or losses) among each arm in PCAWG and simulations. (**c**) Correlation between inferred selection rates and amplification/deletion frequencies for individual chromosome arms. Linear regressions and p-values from Pearson correlation.
(JPG)

**S2 Fig. Inference of chromosome-arm selection rates in Cervix-SCC (PCAWG, $n = 10$ ).** (**a**) Prior distribution (light blue) and posterior distribution (dark blue) from inference with ABC random forest. Broken line represents the mode in the posterior distribution for each parameter. (**b**) Comparison between simulations with fitted parameter (top) and gain/loss frequencies at arm level from TCGA (bottom). The simulations are computed with the posterior modes from (**a**). Spearman's correlation coefficient rho between frequencies of gains (or losses) among each arm in PCAWG and simulations. (**c**) Correlation between inferred selection rates and amplification/deletion frequencies for individual chromosome arms. Linear regressions and p-values from Pearson correlation.
(JPG)

**S3 Fig. Inference of chromosome-arm selection rates in CNS-GBM (PCAWG, $n = 34$ ).** (**a**) Prior distribution (light blue) and posterior distribution (dark blue) from inference with ABC random forest. Broken line represents the mode in the posterior distribution for each parameter. (**b**) Comparison between simulations with fitted parameter (top) and gain/loss frequencies at arm level from TCGA (bottom). The simulations are computed with the posterior modes from (**a**). Spearman's correlation coefficient rho between frequencies of gains (or losses) among each arm in PCAWG and simulations. (**c**) Correlation between inferred selection rates and amplification/deletion frequencies for individual chromosome arms. Linear regressions and p-values from Pearson correlation.
(JPG)

**S4 Fig. Inference of chromosome-arm selection rates in CNS-Oligo (PCAWG, $n = 16$ ).** (**a**) Prior distribution (light blue) and posterior distribution (dark blue) from inference with ABC random forest. Broken line represents the mode in the posterior distribution for each parameter. (**b**) Comparison between simulations with fitted parameter (top) and gain/loss frequencies at arm level from TCGA (bottom). The simulations are computed with the posterior modes from (**a**). Spearman's correlation coefficient rho between frequencies of gains (or losses) among each arm in PCAWG and simulations. (**c**) Correlation between inferred selection rates

and amplification/deletion frequencies for individual chromosome arms. Linear regressions and p-values from Pearson correlation.
(JPG)

**S5 Fig. Inference of chromosome-arm selection rates in ColoRect-AdenoCA (PCAWG, $n = 37$ ). (a)** Prior distribution (light blue) and posterior distribution (dark blue) from inference with ABC random forest. Broken line represents the mode in the posterior distribution for each parameter. **(b)** Comparison between simulations with fitted parameter (top) and gain/loss frequencies at arm level from TCGA (bottom). The simulations are computed with the posterior modes from (**a**). Spearman's correlation coefficient rho between frequencies of gains (or losses) among each arm in PCAWG and simulations. **(c)** Correlation between inferred selection rates and amplification/deletion frequencies for individual chromosome arms. Linear regressions and p-values from Pearson correlation.
(JPG)

**S6 Fig. Inference of chromosome-arm selection rates in Head-SCC (PCAWG, $n = 24$ ). (a)** Prior distribution (light blue) and posterior distribution (dark blue) from inference with ABC random forest. Broken line represents the mode in the posterior distribution for each parameter. **(b)** Comparison between simulations with fitted parameter (top) and gain/loss frequencies at arm level from TCGA (bottom). The simulations are computed with the posterior modes from (**a**). Spearman's correlation coefficient rho between frequencies of gains (or losses) among each arm in PCAWG and simulations. **(c)** Correlation between inferred selection rates and amplification/deletion frequencies for individual chromosome arms. Linear regressions and p-values from Pearson correlation.
(JPG)

**S7 Fig. Inference of chromosome-arm selection rates in Kidney-ChRCC (PCAWG, $n = 33$ ). (a)** Prior distribution (light blue) and posterior distribution (dark blue) from inference with ABC random forest. Broken line represents the mode in the posterior distribution for each parameter. **(b)** Comparison between simulations with fitted parameter (top) and gain/loss frequencies at arm level from TCGA (bottom). The simulations are computed with the posterior modes from (**a**). Spearman's correlation coefficient rho between frequencies of gains (or losses) among each arm in PCAWG and simulations. **(c)** Correlation between inferred selection rates and amplification/deletion frequencies for individual chromosome arms. Linear regressions and p-values from Pearson correlation.
(JPG)

**S8 Fig. Inference of chromosome-arm selection rates in Kidney-RCC (PCAWG, $n = 60$ ). (a)** Prior distribution (light blue) and posterior distribution (dark blue) from inference with ABC random forest. Broken line represents the mode in the posterior distribution for each parameter. **(b)** Comparison between simulations with fitted parameter (top) and gain/loss frequencies at arm level from TCGA (bottom). The simulations are computed with the posterior modes from (**a**). Spearman's correlation coefficient rho between frequencies of gains (or losses) among each arm in PCAWG and simulations. **(c)** Correlation between inferred selection rates and amplification/deletion frequencies for individual chromosome arms. Linear regressions and p-values from Pearson correlation.
(JPG)

**S9 Fig. Inference of chromosome-arm selection rates in Liver-HCC (PCAWG, $n = 31$ ). (a)** Prior distribution (light blue) and posterior distribution (dark blue) from inference with ABC random forest. Broken line represents the mode in the posterior distribution for each

parameter. **(b)** Comparison between simulations with fitted parameter (top) and gain/loss frequencies at arm level from TCGA (bottom). The simulations are computed with the posterior modes from **(a)**. Spearman's correlation coefficient rho between frequencies of gains (or losses) among each arm in PCAWG and simulations. **(c)** Correlation between inferred selection rates and amplification/deletion frequencies for individual chromosome arms. Linear regressions and p-values from Pearson correlation.
(JPG)

**S10 Fig.  Inference of chromosome-arm selection rates in Lung-AdenoCA (PCAWG, $n = 14$ ). (a)** Prior distribution (light blue) and posterior distribution (dark blue) from inference with ABC random forest. Broken line represents the mode in the posterior distribution for each parameter. **(b)** Comparison between simulations with fitted parameter (top) and gain/loss frequencies at arm level from TCGA (bottom). The simulations are computed with the posterior modes from **(a)**. Spearman's correlation coefficient rho between frequencies of gains (or losses) among each arm in PCAWG and simulations. **(c)** Correlation between inferred selection rates and amplification/deletion frequencies for individual chromosome arms. Linear regressions and p-values from Pearson correlation.
(JPG)

**S11 Fig.  Inference of chromosome-arm selection rates in Lung-SCC (PCAWG, $n = 14$ ). (a)** Prior distribution (light blue) and posterior distribution (dark blue) from inference with ABC random forest. Broken line represents the mode in the posterior distribution for each parameter. **(b)** Comparison between simulations with fitted parameter (top) and gain/loss frequencies at arm level from TCGA (bottom). The simulations are computed with the posterior modes from **(a)**. Spearman's correlation coefficient rho between frequencies of gains (or losses) among each arm in PCAWG and simulations. **(c)** Correlation between inferred selection rates and amplification/deletion frequencies for individual chromosome arms. Linear regressions and p-values from Pearson correlation.
(JPG)

**S12 Fig.  Inference of chromosome-arm selection rates in Ovary-AdenoCA (PCAWG, $n = 20$ ). (a)** Prior distribution (light blue) and posterior distribution (dark blue) from inference with ABC random forest. Broken line represents the mode in the posterior distribution for each parameter. **(b)** Comparison between simulations with fitted parameter (top) and gain/loss frequencies at arm level from TCGA (bottom). The simulations are computed with the posterior modes from **(a)**. Spearman's correlation coefficient rho between frequencies of gains (or losses) among each arm in PCAWG and simulations. **(c)** Correlation between inferred selection rates and amplification/deletion frequencies for individual chromosome arms. Linear regressions and p-values from Pearson correlation.
(JPG)

**S13 Fig.  Inference of chromosome-arm selection rates in Prost-AdenoCA (PCAWG, $n = 19$ ). (a)** Prior distribution (light blue) and posterior distribution (dark blue) from inference with ABC random forest. Broken line represents the mode in the posterior distribution for each parameter. **(b)** Comparison between simulations with fitted parameter (top) and gain/loss frequencies at arm level from TCGA (bottom). The simulations are computed with the posterior modes from **(a)**. Spearman's correlation coefficient rho between frequencies of gains (or losses) among each arm in PCAWG and simulations. **(c)** Correlation between inferred selection rates and amplification/deletion frequencies for individual chromosome arms. Linear regressions and p-values from Pearson correlation.
(JPG)

**S14 Fig. Inference of chromosome-arm selection rates in Skin-Melanoma (PCAWG, $n = 16$ ). (a)** Prior distribution (light blue) and posterior distribution (dark blue) from inference with ABC random forest. Broken line represents the mode in the posterior distribution for each parameter. **(b)** Comparison between simulations with fitted parameter (top) and gain/loss frequencies at arm level from TCGA (bottom). The simulations are computed with the posterior modes from (**a**). Spearman's correlation coefficient rho between frequencies of gains (or losses) among each arm in PCAWG and simulations. **(c)** Correlation between inferred selection rates and amplification/deletion frequencies for individual chromosome arms. Linear regressions and p-values from Pearson correlation.
(JPG)

**S15 Fig. Inference of chromosome-arm selection rates in Stomach-AdenoCA (PCAWG, $n = 18$ ). (a)** Prior distribution (light blue) and posterior distribution (dark blue) from inference with ABC random forest. Broken line represents the mode in the posterior distribution for each parameter. **(b)** Comparison between simulations with fitted parameter (top) and gain/loss frequencies at arm level from TCGA (bottom). The simulations are computed with the posterior modes from (**a**). Spearman's correlation coefficient rho between frequencies of gains (or losses) among each arm in PCAWG and simulations. **(c)** Correlation between inferred selection rates and amplification/deletion frequencies for individual chromosome arms. Linear regressions and p-values from Pearson correlation.
(JPG)

**S16 Fig. Inference of chromosome-arm selection rates in Thy-AdenoCA (PCAWG, $n = 47$ ). (a)** Prior distribution (light blue) and posterior distribution (dark blue) from inference with ABC random forest. Broken line represents the mode in the posterior distribution for each parameter. **(b)** Comparison between simulations with fitted parameter (top) and gain/loss frequencies at arm level from TCGA (bottom). The simulations are computed with the posterior modes from (**a**). Spearman's correlation coefficient rho between frequencies of gains (or losses) among each arm in PCAWG and simulations. **(c)** Correlation between inferred selection rates and amplification/deletion frequencies for individual chromosome arms. Linear regressions and p-values from Pearson correlation.
(JPG)

**S17 Fig. Inference of chromosome-arm selection rates in Uterus-AdenoCA (PCAWG, $n = 32$ ). (a)** Prior distribution (light blue) and posterior distribution (dark blue) from inference with ABC random forest. Broken line represents the mode in the posterior distribution for each parameter. **(b)** Comparison between simulations with fitted parameter (top) and gain/loss frequencies at arm level from TCGA (bottom). The simulations are computed with the posterior modes from (**a**). Spearman's correlation coefficient rho between frequencies of gains (or losses) among each arm in PCAWG and simulations. **(c)** Correlation between inferred selection rates and amplification/deletion frequencies for individual chromosome arms. Linear regressions and p-values from Pearson correlation.
(JPG)

**S18 Fig. Inference of chromosome-arm selection rates from pan-cancer TCGA ( $n = 8207$ ). (a)** Prior distribution (light blue) and posterior distribution (dark blue) from inference with ABC random forest. Broken line represents the mode in the posterior distribution for each parameter. **(b)** Comparison between simulations with fitted parameter (top) and gain/loss frequencies at arm level from TCGA (bottom). The simulations are computed with the posterior modes from (**a**). Spearman's correlation coefficient rho between frequencies of gains (or losses) among each arm in TCGA and simulations.

(JPG)

**S19 Fig. Analysis of selection rates for GAIN and LOSS chromosome arms. (a–c)** Impact of varying parameters on clone count **(a)**, average cell fitness **(b)**, and average count of clonal and subclonal missegregations **(c)** (size of circles indicates the total missegregation counts). **(c)** MRCA age and average missegregation counts, grouped based on clonality (clonal/sub-clonal) and type (gain/loss), as selection rates for LOSS arms increase (variables correspond to highlighted segment in **(c)**).
(JPG)

**S20 Fig. Analysis of probability of missegregation and chromosome-arm selection rates.** Impact of varying parameters on average Shannon diversity index **(a)**, average fitness **(b)**, and average ploidy in sample **(c)**.
(JPG)

**S21 Fig. Analysis of growth rate and average cell count.** Impact of varying parameters on clone count **(a)**, and average ploidy in sample **(b)**.
(JPG)

**S22 Fig.** Inference of WGD probability and WGD-aneuploidy rate in individual PCAWG cancer types. Prior distribution (light blue) and posterior distribution (dark blue) from inference with ABC random forest, for Breast-AdenoCA **(a)**, Cervix-SCC **(b)**, CNS-GBM **(c)**, ColoRect-AdenoCA **(d)**, Head-SCC **(e)**, Kidney-ChRCC **(f)**, Kidney-RCC **(g)**, and Liver-HCC **(h)**. Broken line represents the mode in the posterior distribution for each parameter.
(JPG)

**S23 Fig. Inference of WGD probability and WGD-aneuploidy rate in individual PCAWG cancer types. (a–f)** Prior distribution (light blue) and posterior distribution (dark blue) from inference with ABC random forest, for Lung-AdenoCA **(a)**, Lung-SCC **(b)**, Ovary-AdenoCA **(c)**, Skin-Melanoma **(d)**, Stomach-AdenoCA **(e)**, and Uterus-AdenoCA **(f)**. Broken line represents the mode in the posterior distribution for each parameter. **(g and h)** Comparisons between average genomic fraction of monosomy in non-WGD samples and inferred WGD probability **(g)** and WGD-aneuploidy rate **(h)** for each cancer type.
(JPG)

**S24 Fig. Inference of driver gene selection rates in CLLE-ES (PCAWG, $n = 95$). (a)** Ratios of focal amplification lengths over corresponding chromosome arm lengths are fitted with a Beta distribution. **(b)** Prior distribution (light blue) and posterior distribution (dark blue) from inference with ABC random forest. Broken line represents the mode in the posterior distribution for each parameter.
(JPG)

## Author contributions

**Conceptualization:** Khanh N Dinh, Ignacio Vázquez-García, Andrew W. McPherson, Simon Tavaré.

**Data curation:** Ignacio Vázquez-García.

**Formal analysis:** Khanh N Dinh.

**Funding acquisition:** Simon Tavaré.

**Investigation:** Khanh N Dinh.

**Methodology:** Khanh N Dinh, Ignacio Vázquez-García, Andrew W. McPherson.

**Resources:** Ignacio Vázquez-García.

**Software:** Khanh N Dinh, Andrew Chan, Rhea Malhotra, Adam Weiner.

**Supervision:** Andrew W. McPherson, Simon Tavaré.

**Visualization:** Khanh N Dinh, Andrew Chan.

**Writing – original draft:** Khanh N Dinh.

**Writing – review & editing:** Khanh N Dinh, Ignacio Vázquez-García, Andrew W. McPherson, Simon Tavaré.

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
