## [Decision Letter · Decision Letter 0]

26 Aug 2024

Dear Dr. Dinh,

Thank you very much for submitting your manuscript "CINner: modeling and simulation of chromosomal instability in cancer at single-cell resolution" for consideration at PLOS Computational Biology.

As with all papers reviewed by the journal, your manuscript was reviewed by members of the editorial board and by several independent reviewers. In light of the reviews (below this email), we would like to invite the resubmission of a significantly-revised version that takes into account the reviewers' comments.

We cannot make any decision about publication until we have seen the revised manuscript and your response to the reviewers' comments. Your revised manuscript is also likely to be sent to reviewers for further evaluation.

Sincerely,

Simone Zaccaria

Academic Editor

PLOS Computational Biology

Jian Ma

Section Editor

PLOS Computational Biology

Reviewer's Responses to Questions

**Comments to the Authors:**

**Reviewer #1** : CINner: modeling and simulation of chromosomal instability in cancer at single-cell resolution

The manuscript by Dinh and colleagues presents a comprehensive mathematical framework, CINner, for modelling chromosomal instability (CIN) during cancer evolution. The manuscript is well-structured and provides detailed insights into the capabilities of CINner, its applications, and its potential impact on the oncology community.

Major comments:

While the manuscript is detailed, some sections are dense and could benefit from simplification. For example, the section on the different selection models (pages 4-5) could be simplified by summarizing in bullet-point format.

Pg5: “As all three models are defined upon the gene balance in a cell, which is retained after WGD, a WGD cell has the same fitness as its parental cell.” This is quite a big assumption. Should be discussed further as quite a big limitation. What evidence exists that this is a small effect?

Pg5: “Therefore, it is not necessary to simulate single cells in the whole population individually, and instead we focus on clones, defined as groups of cells that have identical CN and mutational characteristics.” This is a sensible approach, but in the introduction you also state CINer “.. can generate tumors with sizes and karyotypes that match observations from DNA-sequencing data from real cancer samples.” I think the statement in the introduction is a bit misleading as you are not directly simulating tumours with realistic sizes.

On a related note, are the population genetic processes at the clonal level guaranteed to recapitulate the dynamics for the full system? I suppose this is captured in the phylogeny part but it is worth discussing and clarifying.

The manuscript could benefit from a more explicit discussion of the limitations of CINner. addressing potential weaknesses and areas for future improvement. Currently only the fixing of the missegregation rate is included.

It would be good to highlight possible ways to get around the nonidentifiability issues, such as using informative priors, hierarchical modelling and so forth.

Minor comments:

Pg11: “region of interest” might be better as “area of interest”

**Reviewer #2** :  Dinh et al. present an ambitious model of chromosomal instability and cancer evolution with an extensive, clear, and detailed supplementary methods section. This work is an exciting approach allowing the incorporation of multiple of mechanisms of genome alteration in simulations and will likely be a useful addition to the literature. However, the manuscript (and online vignettes) require much significant work to improve their clarity before publication to ensure the method is used by others. It would be a real loss if this work is not communicated clearly.  Firstly, simple steps to aid the reviewers such as including line numbers should be undertaken as well as ensuring text in figures is large enough to be read when printed, e.g. in Figures 3 and 4. Furthermore, significant changes including additional schematics and major improvements to existing figures should be made. 

Concerns 1) The authors summarize the literature and contrast their approach with previous models. A main or supplementary table/figure detailing the differences in capabilities between the various previous models and CINer would be a useful addition. This table could also include potentially desirable features of copy number simulations that neither CINer nor other algorithms demonstrate – e.g. allele-specific copy number aberration simulation. 2) Figure 1 could be much improved and/or split into multiple main and supplementary figures. Given the computational audience of the journal, particular care should be taken to clearly demonstrate SNVs and the five distinct CNA mechanisms incorporated by CINer. I would suggest individual simple schematics featuring cartoon chromosomes for each followed by a by visualisations of CINer’s internal representation (current Figure 1b). Care should also be taken regarding the description of each mechanism. The legend for 1b states “Chromosome arm missegregation misplaces a random chromosome arm”, which does not adequately communicate the separate effects on the two daughter cells. Figure 1c is also difficult to interpret with unreadable text and could be much improved. One issue is that the selection model is for chromosome arms, yet the reader is left to identify that arms are represented in the model only by the differences in selection indicated below each heatmap representation of chromosomes 2 and 3.  The legend for Figure 1e is inadequate. What do the vertical dashed arrows indicate. In addition, could the authors describe the various outputs of the method in detail in the supplementary or vignettes?  3) Many design choices are justified by “efficiency” – what are the computational demands of simulating the growth of tumor with this algorithm? What parameter 4) The parameters used by the model and parameter estimation program referenced on page 7 are insufficiently explained in the main text and figures. Schematic figure(s) briefly describing these as well as the analyses and their motivation performed in Figure 2 should be included to aid reader comprehension. 5) The PCAWG and TCGA acronyms are introduced without explanation. In addition, the numbers of cases analyzed should be made clear in supplementary tables and figure legends. The authors could also consider including the datasets and number of samples analyzed in the abstract. 6) The authors should make it clear that tissue and tumor-type specific patterns of aneuploidy are well known and described in the literature with appropriate references. In addition, care should be taken over language. For instance, the sentence “Cancer cells that have amplified GAIN arms are more selective” should be rewritten.  7) Given that the CINer is estimating selection parameters from individual cancer types it would be interesting to see a comparison between the CHARM scores that are estimated from pan-cancer SNV data and their own cancer-type specific scores. For example, for cancer types for which there are sufficient tumors in both PCAWG and the TCGA, could selection parameters be inferred from one dataset and examined on the other as well as being investigated with CHARM scores? One might expect that the cancer-type specific scores inferred by CINer would outperform the pan-cancer CHARM scores. 8) Could the authors clarify their meaning when they say “For instance, the genomes of WGD squamous cell lung carcinoma are significantly altered, but not at a substantially higher level than non-WGD tumours”. How is the word significantly being used here? When examining figure 4a the FGA in the WGD samples looks very different to the non-WGD samples, could the authors comment? 9) On page 12 a sample size is specified at 1000 cells, this seems low – what are the sample sizes used for all other simulations. 10) The authors repeatedly mention the limitations of bulk sequencing data and the possibility of applying CINer to single cell data. I note that the some of authors have previously published papers utilizing single cell datasets. Could the authors consider including a case study of single cell data that could be examined alongside a bulk data from the same cancer type? I understand that this may be beyond the scope of this paper and that the authors may not have bandwidth for such further analysis. 11) In the supplementary page 8, section 1.5.2, $\lambda_d \in (0,\infty)$ (para 1):  Presumably $\lambda_d$ is different for each driver gene? Or not? The sentence is ambiguous.  The next equations only seem to make sense if $\lambda_d \ge 1$. The definitions of $\lambda_{d-WT}$ and $\lambda_{d_MUT}$ seem unmotivated and are unexplained. Would it not be cleaner to have a single style of parametrisation for both, for example the ONCs might have $\lambda_{d-WT}=1$, $\lambda_{d-MUT} = \lambda_d$, if $\lambda_d$ needs to be predefined as greater than 1 for some reason?       Equation (3) on the following page, defining a cell's fitness, seems not to include the non-linear effect on fitness of total loss of both viable alleles of a TSG. Is this intentional?  For example, if one arm containing an ONC/TSG is lost, and the corresponding on the remaining arm is mutated, is the combined model's estimate of fitness realistic? Are the nonlinear fitness effects that occur when LOH exposes a single mutated copy of a gene modelled adequately here? Minor points:  The online Readme has a spelling error “Direct Library Preperation+ (DLP+).” 

**Reviewer #3** : The authors proposed CINner, a framework for modeling chromosomal instability during cancer evolution. I have a few questions regarding the work.

In page 7, parameter calibration should be separately done on the samples with WGD.

In page 19, the authors mentioned that CINner can model rare cancers which are limited by small sample size. But to infer the parameters, does CINner need a large sample size? How can it identify the driver gene given the small sample size?

Fig. 2g is not so clear to see the correlation between WGD proportion and the mean selection rate of the LOSS arm. A dot plot might be better to show the correlation.

It seems that the comparison between the fitted model and the data such as Fig. 2a and other figures used the data being compared to fit the model. This is not fair because it is expected that the model will fit the data if the data is used to come up with the model. Moreover, there is very limited comparisons with other methods. It is mostly just the comparison between the model and the data, whereas the model seems come from the data itself. The cross validation is needed in this case.

The paper compared with only one method, which is citation 22. It shows that it is consistent with the score proposed by citation 22. But is it better than citation 22?

“Importantly, the selection parameter fitting routine only requires a small cohort of cancer samples”. How many cancer samples are needed for the parameter fitting for each cancer type?

Why in 3b the average ploidy never goes up to 4? Is it limited by the cancer types the authors analyzed?

“In previous sections, we estimated cancer type-specific selection parameters and missegregation probabilities in diploid cell populations,” When the ploidy goes up to 3, it is no longer diploid. However, from the figure 3b, the ploidy does go up to 3. I wonder if it is authors’ wrong definition of diploid, or no cell’s ploidy goes beyond 2.

The inference and comparison with PCAWG cohort in Fig. 4d is interesting. I wonder how consistent is counted as satisfactory. For example, Liver-HCC and Cervix-SCC are both not so consistent as other cancer types. Since the authors didn’t compare with any other method, it is hard to give a qualitative conclusion.

**Have the authors made all data and (if applicable) computational code underlying the findings in their manuscript fully available?**

Reviewer #1: Yes

Reviewer #2: Yes

Reviewer #3: Yes

PLOS authors have the option to publish the peer review history of their article (what does this mean? ). If published, this will include your full peer review and any attached files.

**Do you want your identity to be public for this peer review?** For information about this choice, including consent withdrawal, please see our Privacy Policy .

Reviewer #1: No

Reviewer #2: No

Reviewer #3: No
---

## [Decision Letter · Decision Letter 1]

24 Feb 2025

Dear Dr. Dinh,

We are pleased to inform you that your manuscript 'CINner: modeling and simulation of chromosomal instability in cancer at single-cell resolution' has been provisionally accepted for publication in PLOS Computational Biology.

Best regards,

Simone Zaccaria

Academic Editor

PLOS Computational Biology

Jian Ma

Section Editor

PLOS Computational Biology

Reviewer's Responses to Questions

**Comments to the Authors:**

Reviewer #1: Tye authors have satisfied my comments and the paper has been improved.

Reviewer #2: The authors have sufficiently answered all of my comments. I look forward to seeing the article published.

**Have the authors made all data and (if applicable) computational code underlying the findings in their manuscript fully available?**

Reviewer #1: Yes

Reviewer #2: Yes

PLOS authors have the option to publish the peer review history of their article (what does this mean? ). If published, this will include your full peer review and any attached files.

**Do you want your identity to be public for this peer review?** For information about this choice, including consent withdrawal, please see our Privacy Policy .

Reviewer #1: No

Reviewer #2: No

---

## [Editor Report · Acceptance letter]

PCOMPBIOL-D-24-00930R1

CINner: modeling and simulation of chromosomal instability in cancer at single-cell resolution

Dear Dr Dinh,

I am pleased to inform you that your manuscript has been formally accepted for publication in PLOS Computational Biology. Your manuscript is now with our production department and you will be notified of the publication date in due course.

With kind regards,

Anita Estes
